# A multisensor high-temperature signaling framework for triggering daytime thermomorphogenesis in Arabidopsis

De Fan [1], Wei Hu [2], Nan Xu[1], Ethan R. Seto[1], John Clark Lagarias [2], Xuemei Chen [1,3,4] ✉ & Meng Chen [1] ✉

The phytochrome B (phyB) photoreceptor and EARLY FLOWERING 3 (ELF3) are two major plant thermosensors that monitor high temperatures primarily at night. However, high temperatures naturally occur during the daytime; the mechanism of daytime thermosensing and whether these thermosensors can also operate under intense sunlight remain ambiguous. Here, we show that phyB plays a substantial role in daytime thermosensing in Arabidopsis, and its thermosensing function becomes negligible only when the red light intensity reaches 50 $\mu mol\,m^{-2}\,s^{-1}$. Leveraging this restrictive condition for phyB thermosensing, we reveal that triggering daytime thermomorphogenesis requires two additional thermosensory pathways. High temperatures induce starch breakdown in chloroplasts and the production of sucrose, which stabilizes the central thermal regulator PHYTOCHROME-INTERACTING FACTOR 4 (PIF4) by antagonizing phyB-dependent PIF4 degradation. In parallel, high temperatures release the inhibition of *PIF4* transcription and PIF4 activity by ELF3. Thus, our study elucidates a multisensor high-temperature signaling framework for understanding diverse thermo-inducible plant behaviors in daylight.

Sensing ambient temperature changes is essential for plant survival. While extreme heat and cold elicit stress tolerance responses, a moderate increase in ambient temperature can dramatically alter plant development, growth, metabolism, and immunity, a process called thermomorphogenesis[1–4]. A deeper understanding of the thermosensing mechanisms underpinning thermomorphogenesis is pivotal for understanding the impact of diverse geographical environments on plant species distribution and agricultural productivity, as well as for optimizing indoor conditions to improve crop yield in vertical farming.

Thermomorphogenesis is best studied in the seedling development of *Arabidopsis thaliana* (Arabidopsis), where high temperatures induce the rapid growth of the embryonic stem (hypocotyl)[1–4]. Hypocotyl growth is dynamically and precisely controlled by a variety of environmental stimuli and endogenous phytohormones via the regulation of the transcription, stability, and activity of a small family of nodal basic helix-loop-helix transcription factors called PHYTOCHROME-INTERACTING FACTORs (PIFs)[5,6]. Five PIFs, including PIF1, PIF3, PIF4, PIF5, and PIF7, can enhance hypocotyl growth by activating growth-relevant genes, such as genes involved in the biosynthesis and signaling of the growth-promoting hormone auxin[7–11]. Thermo-inducible hypocotyl growth is mediated primarily by the central thermal regulator PIF4, along with PIF5 and PIF7[12–17]. High temperatures enhance both *PIF4* transcription and PIF4 stability, thereby leading to the rapid accumulation of PIF4 protein—a critical step in initiating thermomorphogenesis.

Three thermosensors have been reported, and they sense temperature through distinct mechanisms that converge to control the abundance of PIF4 and PIF7[14,18–20]. First, the phytochrome B (phyB)

[1]Department of Botany and Plant Sciences, Institute for Integrative Genome Biology, University of California, Riverside, CA, USA. [2]Department of Molecular and Cellular Biology, University of California, Davis, CA, USA. [3]State Key Laboratory for Gene Function and Modulation Research, Peking-Tsinghua Joint Center for Life Sciences, College of Life Sciences, Peking University, Beijing, China. [4]Beijing Advanced Center of RNA Biology (BEACON), Peking University, Beijing, China. ✉e-mail: xuemei.chen@pku.edu.cn; meng.chen@ucr.edu

photoreceptor is considered a thermosensor because high temperatures destabilize its active form[18,19]. PhyB senses red (R) and far-red (FR) light through photo-interconversion between an inactive R-light-absorbing Pr form and an active FR-light-absorbing Pfr form[21,22]. Temperature increases between 10 and 30 °C destabilize Pfr by accelerating the light-independent thermal reversion of Pfr back to the inactive Pr[19]. Because photoactivated phyB binds PIF4 directly to promote PIF4 degradation, phyB inactivation by high temperatures promotes PIF4 accumulation[15,16,23–25]. Second, high temperatures are sensed via the thermo-inducible phase separation of the prion-like-domain-containing protein EARLY FLOWERING 3 (ELF3)[20,26–28]. ELF3 is a component of the Evening Complex−a transcriptional repressor complex that also contains EARLY FLOWERING 4 (ELF4) and the DNA-binding protein LUX ARRHYTHMO (LUX)−which represses *PIF4* transcription[29]. ELF3 condensation reduces the binding of the Evening Complex to the *PIF4* promoter, thereby de-repressing *PIF4* transcription at high temperatures[20,26,30]. Third, the thermosensitive unfolding of a hairpin structure within the 5′ untranslated region of the *PIF7* transcript enhances PIF7 translation and therefore can contribute to thermomorphogenesis[14].

In current thermosensing models, the two major thermosensors, phyB and ELF3, function primarily at night or in darkness[18,19,30]. However, in nature, plants are more likely to encounter high temperatures during the daytime under intense sunlight. Daytime thermosensing is expected to differ from nighttime thermosensing for the following reasons[16,25]. First, theoretically, the phyB-dependent thermosensing mechanism can operate only under dim light and at night, because phyB's thermal reversion should be rendered negligible due to rapid photoactivation in strong daylight conditions[3,19]. Second, in contrast to night conditions, phyB is photoactivated by daylight and thus it stays active to continually promote PIF4 degradation. Therefore, daytime thermo-responsive hypocotyl growth requires an additional mechanism to stabilize PIF4 by antagonizing phyB-mediated PIF4 degradation[16,25]. Third, because hypocotyl growth is gated by the circadian clock and partitioned to different times in short-day (SD, as in winter) and long-day (LD, as in summer) conditions, daytime and nighttime thermosensing mechanisms are assessed under distinct assay conditions. Hypocotyl elongation occurs mainly at the end of the night in SD conditions but peaks during the daytime in LD or continuous light conditions[31–33]. Thus, while nighttime thermosensing is best assessed in SD conditions, daytime thermosensing is best quantified in LD or continuous light conditions[34]. Since the Evening Complex represses *PIF4* transcription specifically in the early night under SD conditions[29], ELF3 is thought to sense temperature only at night[20,30,35]. Supporting this idea, the *elf3* mutant is defective in thermo-inducible hypocotyl growth only in SD conditions but can still respond to high temperatures in LD and continuous light conditions[34,35]. As such, the daytime thermosensing mechanism remains poorly understood.

Overwhelming evidence demonstrates that plants do respond to high temperatures during the daytime under LD and continuous light conditions[14,16,35]. Thermo-inducible hypocotyl elongation is repressed by the blue light photoreceptor cryptochrome 1 under certain conditions[16,36]. This blue-light effect can be readily circumvented by assessing daytime thermosensing under monochromatic R light[16,34]. Here, we used thermoresponsive hypocotyl growth as a model to examine whether phyB's thermal reversion plays a role in daytime thermosensing in continuous R light conditions and whether other thermosensing mechanisms exist specifically for thermosensing in daylight. We show that the thermal reversion of phyB plays a substantial role in thermosensing in the light and that the thermosensing role of phyB becomes negligible only when the R light intensity reaches 50 µmol m$^{-2}$ s$^{-1}$. Leveraging this restrictive condition for phyB thermosensing allowed us to identify and dissect two additional thermosensory pathways required for initiating daytime thermomorphogenesis. High temperatures trigger the breakdown of starch

and the production of sucrose, which stabilizes PIF4 by antagonizing phyB-mediated PIF4 degradation. In parallel, high temperatures also release the inhibition of *PIF4* transcription and PIF4 activity by ELF3. The concerted actions of these chloroplast-sucrose-mediated and ELF3-dependent pathways converge to enable PIF4 to initiate daytime thermomorphogenesis. Our results elucidate a multisensor high-temperature signaling framework for understanding diverse thermo-responsive plant behaviors in daylight.

## Results

### Dissecting phyB-independent thermosensing in the light

The phyB thermal-reversion-dependent thermosensing mechanism operates across an ambient temperature range of 10−30 °C[3,19]. Because the photoactivation of phyB counteracts its thermal reversion, the effect of phyB thermosensing is expected to be repressed under intense light, casting doubts on the significance of phyB-dependent thermosensing in daylight conditions[3,19]. However, the threshold of R light intensity that can abolish the effect of phyB thermosensing has not been determined. To dissect the role of phyB thermal reversion in daytime thermosensing, we used thermoresponsive hypocotyl growth in Arabidopsis to examine the effect of R light intensity on thermomorphogenesis. To that end, we generated temperature-response curves for hypocotyl growth in Col-0 between 12 and 27 °C under R light intensities ranging from 0.1 to 50 µmol m$^{-2}$ s$^{-1}$. Surprisingly, the temperature-response curves revealed two distinct temperature responses: a light-repressible response at the low-temperature range between 12 and 21 °C (Fig. 1a, the blue-shaded temperature range) and a non-light-repressible response at the high-temperature range between 21 and 27 °C (Fig. 1a, the pink-shaded temperature range). These results suggest that phyB thermal reversion is the dominant thermosensing mechanism between 12 and 21 °C, which we refer to as a phyB-dependent response (Fig. 1a). Among the light intensities tested, only 50 µmol m$^{-2}$ s$^{-1}$ R light almost completely abolished the phyB-dependent response between 12 and 21 °C (Fig. 1a), indicating that the threshold of R light intensity required for repressing the phyB-dependent response is at least 50 µmol m$^{-2}$ s$^{-1}$, which is considered very strong light. These results indicate that phyB thermal reversion can play a significant role in thermosensing in a wide range of light conditions, likely also including the temperature range between 21 and 27 °C[3,19].

This strong 50 µmol m$^{-2}$ s$^{-1}$ R light condition represented a restrictive condition for the effect of phyB thermosensing and therefore allowed us to dissect the thermosensing mechanisms independent of phyB thermal reversion. Notably, the response at the high-temperature range between 21 and 27 °C−which defines thermomorphogenesis in Arabidopsis[1,3,4]−was light-irrepressible and persisted even under 50 µmol m$^{-2}$ s$^{-1}$ R light (Fig. 1a). These results indicate that thermo-inducible hypocotyl growth must also be mediated by phyB-independent thermosensing mechanisms. Unlike the graded phyB-dependent response operating from 10 to 30 °C[3,19], the phyB-independent thermosensing mechanism acts like a switch triggered only by the temperature increase from 21 to 27 °C (Fig. 1a). Therefore, thermosensing between 21 and 27 °C must be mediated by both phyB-dependent and phyB-independent mechanisms, which we refer to as a multisensor-dependent mechanism (Fig. 1a). To highlight the difference between the phyB-dependent and multisensor-dependent temperature responses, we plotted the temperature-induced hypocotyl growth responses at the low (from 12 to 21 °C) and high (from 21 to 27 °C) temperature ranges against light intensity. As shown in Fig. 1b, while the phyB-dependent temperature response was repressed by increases in light intensity, the multisensor-dependent temperature response was enhanced in strong light conditions (Fig. 1b).

To further confirm the role of phyB thermal reversion in the phyB-dependent and multisensor-dependent responses, we examined the two responses in the *YHB$^g$* line, which carries the genomic *PHYB* DNA encoding a constitutively active phyB$^{Y276H}$ mutant that lacks thermal reversion[37–40]. Compared with the L*er* wild-type, *YHB$^g$* seedlings

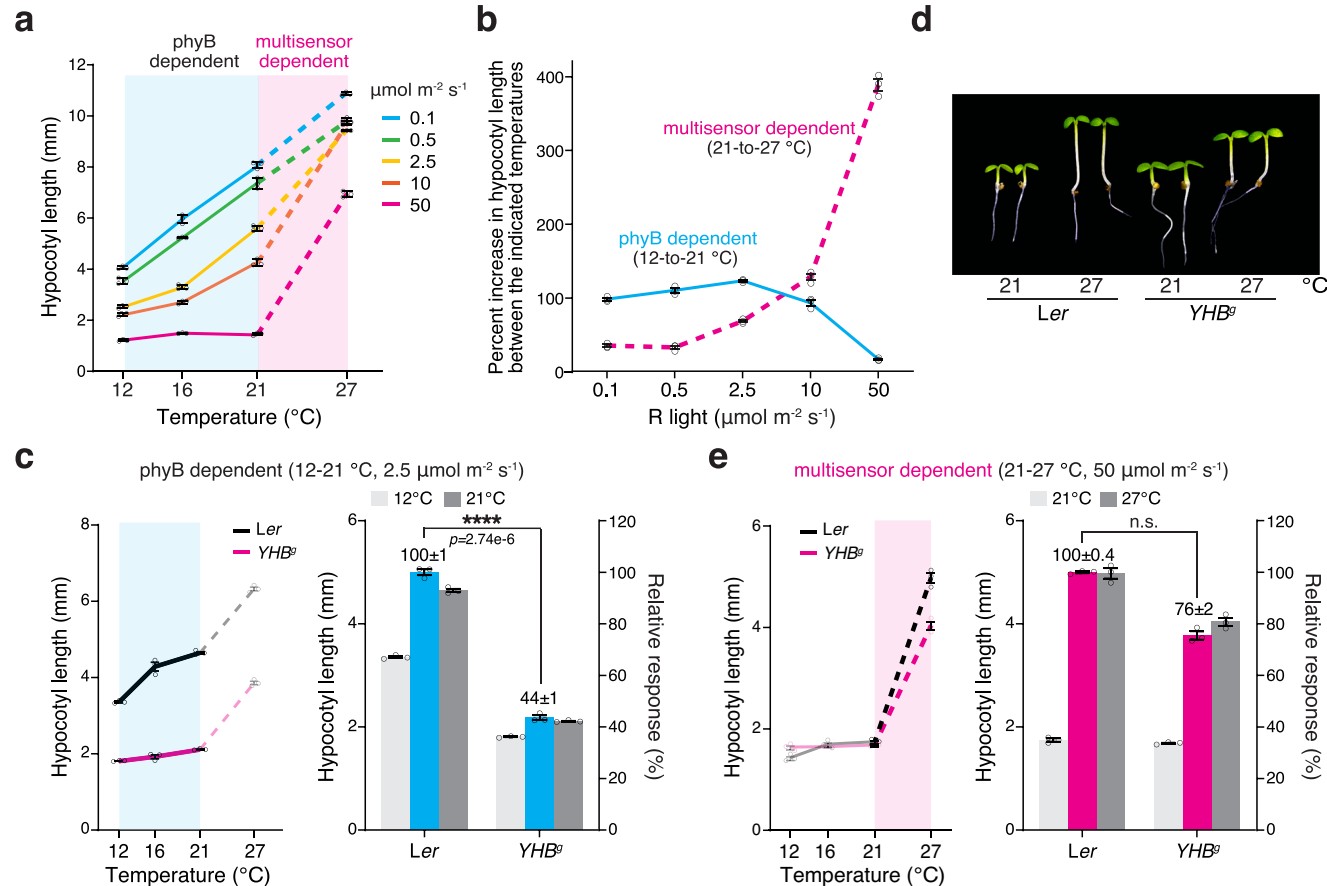

**Fig. 1 | Dissecting phyB-dependent and phyB-independent thermosensing in the light. a** Temperature-response curves of hypocotyl elongation under various intensities of R light reveal distinct thermosensing mechanisms in the light: a mechanism depending solely on phyB thermal reversion (phyB dependent) at low temperatures and a mechanism also requiring thermosensing independently of phyB thermal reversion (multisensor dependent) at high temperatures. Hypocotyl lengths were measured in 4-d-old Col-0 seedlings grown at 12 °C, 16 °C, 21 °C, and 27 °C under a range of continuous R light intensities from 0.1 to 50 µmol m⁻² s⁻¹. The blue and pink shades highlight the temperature responses at the low-temperature range between 12 and 21 °C (solid lines) and the high-temperature range between 21 and 27 °C (dashed lines), respectively. **b** Distinct effects of light intensity on the low- and high-temperature responses. The phyB-dependent low-temperature (12–21 °C) and multisensor-dependent high-temperature (21–27 °C) responses were quantified as the percent increase in hypocotyl length in the respective temperature ranges and plotted against light intensity. **c** *YHBᵍ* seedlings lack the phyB-dependent low-temperature response. The left panel shows the temperature-response curve of 4-d-old L*er* and *YHBᵍ* seedlings grown under 2.5 µmol m⁻² s⁻¹ R light. The blue shade highlights the phyB-dependent response between 12 and

21 °C. The right panel shows the hypocotyl lengths of L*er* and *YHBᵍ* seedlings at 12 and 21 °C. The blue bars show the relative response, which is defined as the hypocotyl response at 21 °C in *YHBᵍ* relative to that in L*er* (set at 100%). **d** Images of 4-d-old L*er* and *YHBᵍ* seedlings grown under 50 µmol m⁻² s⁻¹ R light at either 21 or 27 °C. **e** *YHBᵍ* seedlings retained the multisensor-dependent high-temperature response. The left panel shows the temperature-response curve of 4-d-old L*er* and *YHBᵍ* seedlings grown under 50 µmol m⁻² s⁻¹ R light. The pink shade highlights the multisensor-dependent response between 21 and 27 °C. The right panel shows the hypocotyl lengths of L*er* and *YHBᵍ* seedlings at 21 and 27 °C. The magenta bars show the relative response, which is defined as the hypocotyl response at 27 °C in *YHBᵍ* relative to that in L*er* (set at 100%). For (**a**–**c** and **e**), the error bars represent the s.e. (*n* = three biological replicates), and the centers of the error bars indicate the mean. For (**c** and **e**), a significant difference in the relative response between L*er* and *YHBᵍ* was defined as a greater than twofold and statistically significant change (*p* < 0.05) in the temperature response based on a two-tailed Student's *t*-test (**** indicates *p* < 0.0001) and otherwise was defined as no significant difference (n.s.). The underlying source data for the hypocotyl measurements in (**a**–**c** and **e**) are provided in the Source Data file.

exhibited a significantly reduced response between 12 and 21 °C but a similar response between 21 and 27 °C (Fig. 1c–e), demonstrating that the response in the low-temperature range depends solely on phyB thermal reversion and that the high-temperature response requires a phyB thermal-reversion-independent thermosensing mechanism. The *YHBᵍ* line showed a significantly stronger thermal response than a previously reported line called *YHB*, which expresses phyB^Y276H under the constitutive 35S promoter[41,42]. This discrepancy is likely due to the higher level of phyB^Y276H in *YHB*, which could ectopically repress thermoresponsive hypocotyl elongation (Supplementary Fig. 1).

## Chloroplasts and sucrose are required for the initiation of thermomorphogenesis

We leveraged the 50 µmol m⁻² s⁻¹ R light condition to investigate the phyB thermal-reversion-independent thermosensing mechanism

between 21 and 27 °C. Although phyB thermal reversion is negligible for thermosensing under intense R light conditions, the high-temperature response still relies on phyB signaling to repress hypocotyl growth, as the null *phyB-9* mutant fails to respond to high temperatures and has a long hypocotyl at both 21 and 27 °C[16]. Therefore, in the multisensor mechanism, temperature changes could be sensed by either a downstream component of phyB signaling or a parallel pathway that intersects with phyB signaling. To distinguish between these two possibilities, we tested whether *YHBᵍ* could respond to high temperatures in darkness, where only phyB signaling is turned on. To our surprise, dark-grown *YHBᵍ* lacked the high-temperature response (Fig. 2a, b). Therefore, high-temperature is most likely not sensed by phyB signaling per se but rather via a parallel pathway that antagonizes phyB signaling, and this thermosensing mechanism can operate only in seedlings grown in the light.

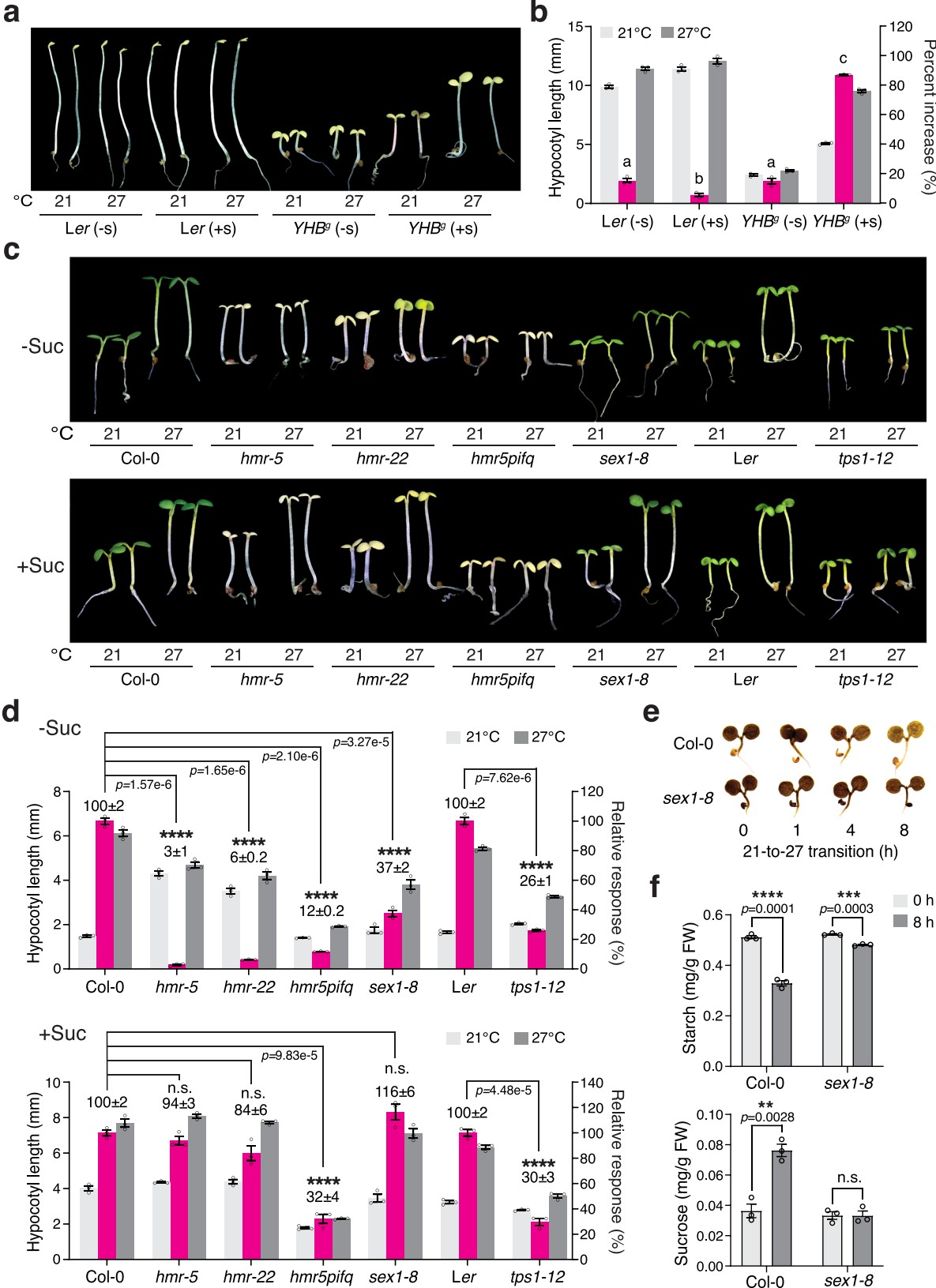

One of the distinguishing characteristics of light-grown seedlings compared to dark-grown etiolated seedlings is the biogenesis of photosynthetically active chloroplasts. Intriguingly, supplying sucrose—the mobile sugar product of photosynthesis—to the growth media restored the thermo-inducible hypocotyl growth of *YHB^g* seedlings in the dark (Fig. 2a, b), suggesting that thermomorphogenesis

may rely on chloroplasts. We previously reported that the chloroplast-deficient *hemera* (*hmr*) mutant—which disrupts the plastid-encoded RNA polymerase and therefore fails to express plastid-encoded photosynthesis genes required for carbon assimilation and sugar production—is impaired in thermomorphogenesis under intense R light conditions[16]. Strikingly, supplementing with sucrose also restored the

**Fig. 2 | Thermo-inducible starch breakdown in chloroplasts is required for thermomorphogenesis in the light. a** Images of 4-d-old L*er* and *YHB^g* seedlings grown in the dark at either 21 or 27 °C without (-s) or with (+s) sucrose. **b** Hypocotyl length measurements showing the thermoresponsive hypocotyl growth of the L*er* and *YHB^g* seedlings in (**a**). The magenta bars show the percent increase in hypocotyl length at 27 °C compared to 21 °C. Different letters denote statistically significant differences in the change in hypocotyl length (one-way ANOVA, Tukey's HSD, $p < 0.05$, $n = 3$ biological replicates). **c** Images of 4-d-old Col-0, *hmr-5*, *hmr-22*, *hmr-5pifq*, *sex1-8*, L*er*, and *tps1-12* seedlings grown under 50 μmol m$^{-2}$ s$^{-1}$ R light at either 21 or 27 °C without or with sucrose. **d** Hypocotyl length measurements of the genotypes described in (**c**). The magenta bars show the relative response, which is defined as the hypocotyl response of a mutant at 27 °C relative to that of the corresponding wild-type (set at 100%). A significant difference in the relative response between the indicated mutants and the wild-type was defined as a greater

than twofold and statistically significant change based on a two-tailed Student's *t*-test (** $p < 0.01$, *** $p < 0.001$, **** $p < 0.0001$), and otherwise was defined as no significant difference (n.s.). **e** Iodine staining showing the starch contents of 4-d-old Col-0 and *sex1-8* seedlings grown under 50 μmol m$^{-2}$ s$^{-1}$ R light at the indicated time points during the 21-27 °C transition. **f** Quantification of the starch (left panel) and sucrose (right panel) levels in Col-0 and *sex1-8* seedlings at the 0 and 8 h time points during the 21–27 °C transition shown in (**e**). A significant difference between the 0 and 8 h samples was calculated using a two-tailed Student's *t*-test (** $p < 0.01$, *** $p < 0.001$, **** $p < 0.0001$). For (**b**, **d**, and **f**), all error bars represent the s.e. ($n$ = three biological replicates), and the centers of the error bars indicate the mean. The underlying source data for the hypocotyl measurements in (**b** and **d**) and for the starch and sucrose quantifications in (**f**) are provided in the Source Data file.

thermoresponsive hypocotyl growth in the null *hmr-5* and weak *hmr-22* mutants (Fig. 2c, d). These results indicate that chloroplasts and sucrose production are required for thermosensing under strong light conditions. *hmr*'s defect in the thermal response is caused by the inability to antagonize the phyB-mediated inhibition of hypocotyl growth, as opposed to a lack of energy, because the *hmr-2/phyB-9* mutant can grow as tall as *phyB-9* in R light without exogenous sugar[43]. Therefore, instead of providing the necessary energy for growth, sucrose restored *hmr*'s thermoresponsive hypocotyl growth most likely via the action of sugar signaling.

A large proportion of the fixed carbon produced by photosynthesis during the day is retained in the source tissue, such as the cotyledons, as transitory starch in chloroplasts[44,45]. Starch is mobilized to produce sucrose for export to the sink tissues at night[46,47]. We tested whether high temperatures could induce starch degradation in the light. To our surprise, under intense R light, a temperature increase from 21 to 27 °C triggered a rapid reduction in starch content and, concomitantly, a significant increase in the level of sucrose in Col-0 (Fig. 2e, f). The initiation of starch degradation in the dark and the cold requires a starch-bound α-glucan water dikinase named GWD or STARCH EXCESS 1 (SEX1), which is involved in starch phosphorylation/dephosphorylation[48–50]. The *sex1* mutant is partially defective in starch breakdown, resulting in a very strong starch-excess phenotype[44,47,49,51]. The null *sex1-8* mutant also showed a marked reduction in starch degradation and sucrose production during the 21–27 °C transition (Fig. 2e, f). Consistent with the hypothesis that thermo-inducible starch degradation in chloroplasts is required for thermomorphogenesis, the *sex1-8* mutant was significantly impaired in thermoresponsive hypocotyl growth, and this defect could be reversed by exogenous sucrose (Fig. 2c, d). Sucrose signaling is mediated by the metabolite signal trehalose-6-phosphate (Tre6P), which is made from trehalose by trehalose-6-phosphate synthase (TPS)[52]. Reducing Tre6P biosynthesis in the *tps1-12* mutant attenuated thermoresponsive hypocotyl growth, and this defect could not be restored by exogenous sucrose (Fig. 2c, d). These results are in agreement with previous studies showing that thermomorphogenesis requires Tre6P-mediated sucrose signaling[53]. Together, our results support the conclusion that chloroplasts play an essential role in the phyB thermal-reversion-independent thermosensory mechanism. The thermo-inducible starch degradation in chloroplasts and subsequent sucrose production and signaling are critical steps in initiating thermoresponsive hypocotyl growth in the light.

### Sucrose signaling promotes PIF4 accumulation

We next investigated how sucrose promotes thermomorphogenesis. While exogenous sucrose could restore thermoresponsive hypocotyl growth in *hmr-5*, it could not rescue the *hmr-5/pifq* mutant, which harbors quadruple *pif1*, *pif3*, *pif4*, and *pif5* mutations (*pifq*) (Fig. 2c,d), indicating that sucrose exerts its function through PIFs. Among the four PIFs, PIF4 plays a prominent role in initiating

thermomorphogenesis by activating genes involved in auxin biosynthesis and signaling[1–4]. Increasing the ambient temperature from 21 to 27 °C triggered the rapid accumulation of PIF4 in Col-0 under intense R light (Fig. 3a, b)[1–4]. Consistent with our previous reports, *hmr* mutants failed to accumulate PIF4 at high temperatures (Fig. 3a, b)[16]. Interestingly, exogenous sucrose greatly enhanced the steady-state levels of PIF4 in the null *hmr-5* mutant at both 21 and 27 °C and in the weak *hmr-22* allele during the 21–27 °C transition (Fig. 3a, b), indicating that *hmr*'s defect in PIF4 accumulation is attributable to sugar deficiency. The fact that exogenous sucrose could stimulate PIF4 accumulation in Col-0, *hmr-5*, and *hmr-22* at 21 °C supports the idea that sucrose production alone is sufficient to trigger PIF4 accumulation (Fig. 3a, b). This thermo- and sucrose-inducible PIF4 accumulation was lost in the *phyB-9* mutant, in which PIF4 accumulated to high levels regardless of temperature or sucrose treatment (Fig. 3a), indicating that sucrose promotes PIF4 accumulation by antagonizing phyB-mediated PIF4 degradation. To determine whether thermoresponsive PIF4 accumulation depends on starch degradation and sucrose signaling, we examined PIF4 levels in *sex1-8* and *tps1-12*. Although the steady-state levels of PIF4 in *sex1-8* at 21 and 27 °C were similar to those in Col-0, high-temperature-induced PIF4 accumulation during the 21–27 °C transition was delayed in *sex1-8*, and the thermal induction of *YUC8* and *IAA19*—two PIF4 target genes involved in auxin biosynthesis and signaling—was significantly impaired (Fig. 3c–e). In contrast, the *PIF4* transcript in *sex1-8* could still be elevated by high temperatures, similar to that in Col-0 (Fig. 3e), indicating that starch degradation promotes PIF4 accumulation posttranscriptionally. The thermoresponsive accumulation of PIF4 and induction of *YUC8* and *IAA19* were completely abolished in *tps1-12*, whereas the high-temperature-induced increase in *PIF4* transcripts remained the same as in L*er* (Fig. 3f–h). Together, these results support the conclusion that sucrose signaling promotes PIF4 accumulation by antagonizing phyB-mediated PIF4 degradation.

### Dual thermosensory pathways license PIF4 activity

Although sucrose could dramatically enhance PIF4 levels in the *hmr* mutants at 21 °C (Fig. 3a, b), it was puzzling that PIF4 accumulation alone was insufficient to trigger hypocotyl growth; that is, even in the presence of sucrose, *hmr-5* and *hmr-22* still required elevated temperatures to trigger hypocotyl growth (Fig. 2c, d). Similarly, exogenous sucrose could not elicit a complete hypocotyl growth response in Col-0, *sex1-8*, *YHB^g* at 21 °C to the same extent as it would with a 27 °C treatment (Fig. 2a–d). These observations indicate that in addition to the chloroplast-sucrose-mediated thermosensory pathway, there is yet another separate thermosensor (referred to as Sensor 2). Sensor 2 must inhibit hypocotyl growth at 21 °C, and its growth-inhibitory function is released at 27 °C. Sensor 2 alone also cannot trigger hypocotyl growth because, even at 27 °C, both light-grown *hmr* mutants and dark-grown *YHB^g* still required sucrose to stimulate hypocotyl growth (Fig. 2a–d). Therefore, thermo-inducible hypocotyl

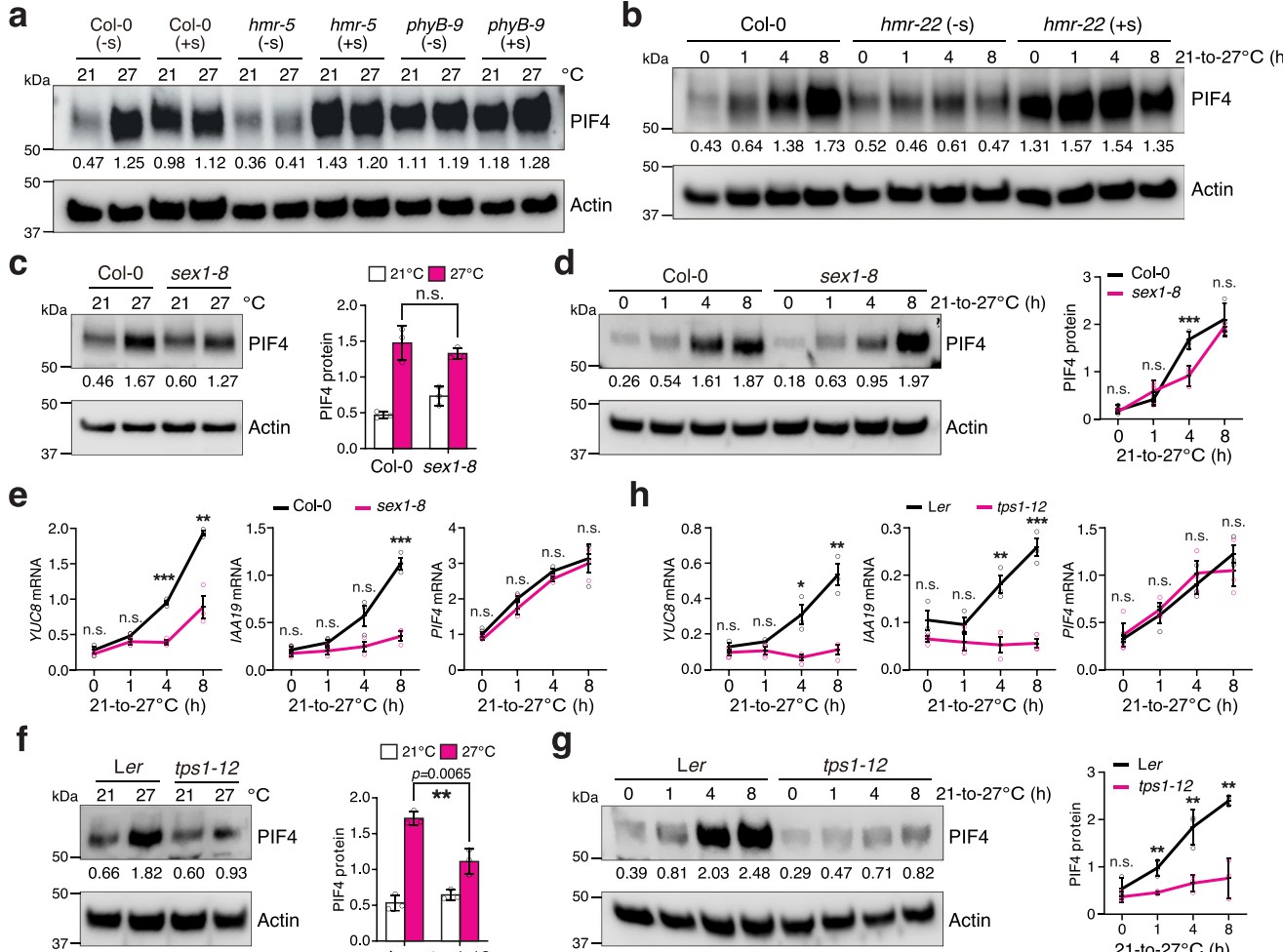

**Fig. 3 | Sucrose promotes PIF4 accumulation. a** Immunoblots showing the PIF4 levels in 4-d-old Col-0, *hmr-5*, and *phyB-9* seedlings grown under 50 µmol m⁻² s⁻¹ R light at either 21 or 27 °C without (-s) or with (+s) sucrose. **b** Immunoblots showing the PIF4 levels in 4-d-old Col-0 and *hmr-22* seedlings under 50 µmol m⁻² s⁻¹ R light without or with sucrose treatment during the 21–27 °C transition. **c** Immunoblots showing the PIF4 levels in 4-d-old Col-0 and *sex1-8* seedlings grown under 50 µmol m⁻² s⁻¹ R light at either 21 or 27 °C. **d** Immunoblots showing the PIF4 levels in 4-d-old Col-0 and *sex1-8* seedlings grown under 50 µmol m⁻² s⁻¹ R light during the 21–27 °C transition. **e** Quantitative real-time PCR results showing the transcript levels of *YUC8*, *IAA19*, and *PIF4* in Col-0 and *sex1-8* seedlings during the 21–27 °C transition as described in (**d**). **f** Immunoblots showing the PIF4 levels in 4-d-old L*er* and *tps1-12* seedlings grown under 50 µmol m⁻² s⁻¹ R light at either 21 or 27 °C. **g** Immunoblots showing the PIF4 levels in 4-d-old L*er* and *tps1-12* seedlings grown under 50 µmol m⁻² s⁻¹ R light during the 21–27 °C transition. **h** Quantitative real-time PCR (qRT-PCR) results showing the transcript levels of *YUC8*, *IAA19*, and *PIF4* in L*er* and *tps1-12* seedlings during the 21–27 °C transition as described in (**g**). For

(**a–d**, **f**, and **g**), actin was used as the loading control. The relative PIF4 levels normalized to actin are shown. The immunoblot experiments were independently repeated three times with similar results. Error bars for the protein quantifications in (**c**, **d**, **f**, and **g**) represent the s.d. (*n* = three biological replicates), and the centers of the error bars indicate the mean. Significant differences between the PIF4 protein levels in Col-0 and the mutants were calculated using two-tailed Student's *t*-tests (** *p* < 0.01, *** *p* < 0.001); n.s. stands for no significant difference. For (**e** and **h**), the transcript levels of the indicated genes were calculated relative to those of *PP2A*. Error bars represent the s.e. (*n* = three biological replicates), and the centers of the error bars indicate the mean. Significant differences in the transcript levels of the indicated genes between Col-0 and the mutants were calculated using two-tailed Student's *t*-tests (* *p* < 0.05, ** *p* < 0.01, *** *p* < 0.001); n.s. stands for no significant difference. The underlying source data for the immunoblots in (**a–d**, **f**, and **g**) and the qRT-PCR results in (**e** and **h**) are provided in the Source Data file.

growth must be triggered by the concerted actions of the growth-promoting function of the chloroplast-sucrose-mediated thermosensory pathway (to stabilize PIF4) and the derepression of the growth-inhibitory function of Sensor 2.

To determine the combined actions of sucrose and Sensor 2, we performed a genome-wide transcriptome analysis to identify the thermo-inducible genes that depend on both sucrose and Sensor 2. We leveraged the *hmr-5* mutant in this analysis because its chloroplast defect provides a clean sucrose-deficient condition that can be compared with the sucrose supplement condition to dissect the effect of sucrose. By comparing the transcriptomic profiles of *hmr-5* seedlings grown at 27 °C with sucrose (*hmr-5_*27S) to those without sucrose (*hmr-5_*27)—in both conditions, the growth-inhibitory function of Sensor 2

was released—we identified 3561 sucrose-induced genes (sucrose-induced) (Fig. 4a and Supplementary Data 1). By comparing the transcriptomic profiles of *hmr-5* seedlings grown with sucrose at 27 °C (*hmr-5_*27S) to those growth with sucrose at 21 °C (*hmr-5_*21S)—in both conditions, PIF4 accumulated to high levels due to sucrose—we identified 848 genes induced by releasing the growth-inhibitory function of Sensor 2 (sensor2-induced) (Fig. 4a and Supplementary Data 1). Sucrose and Sensor 2 induced 224 common genes, among which 173 depended on PIFs for their induction as they were not thermally induced in *hmr-5pifq* (Fig. 4a). These 173 genes were named dual-sensor-induced PIF-dependent genes, as their induction requires both sucrose and Sensor 2 (Fig. 4a, b and Supplementary Data 1). The top Gene Ontology (GO) category enriched in the dual-sensor-induced and PIF-dependent genes

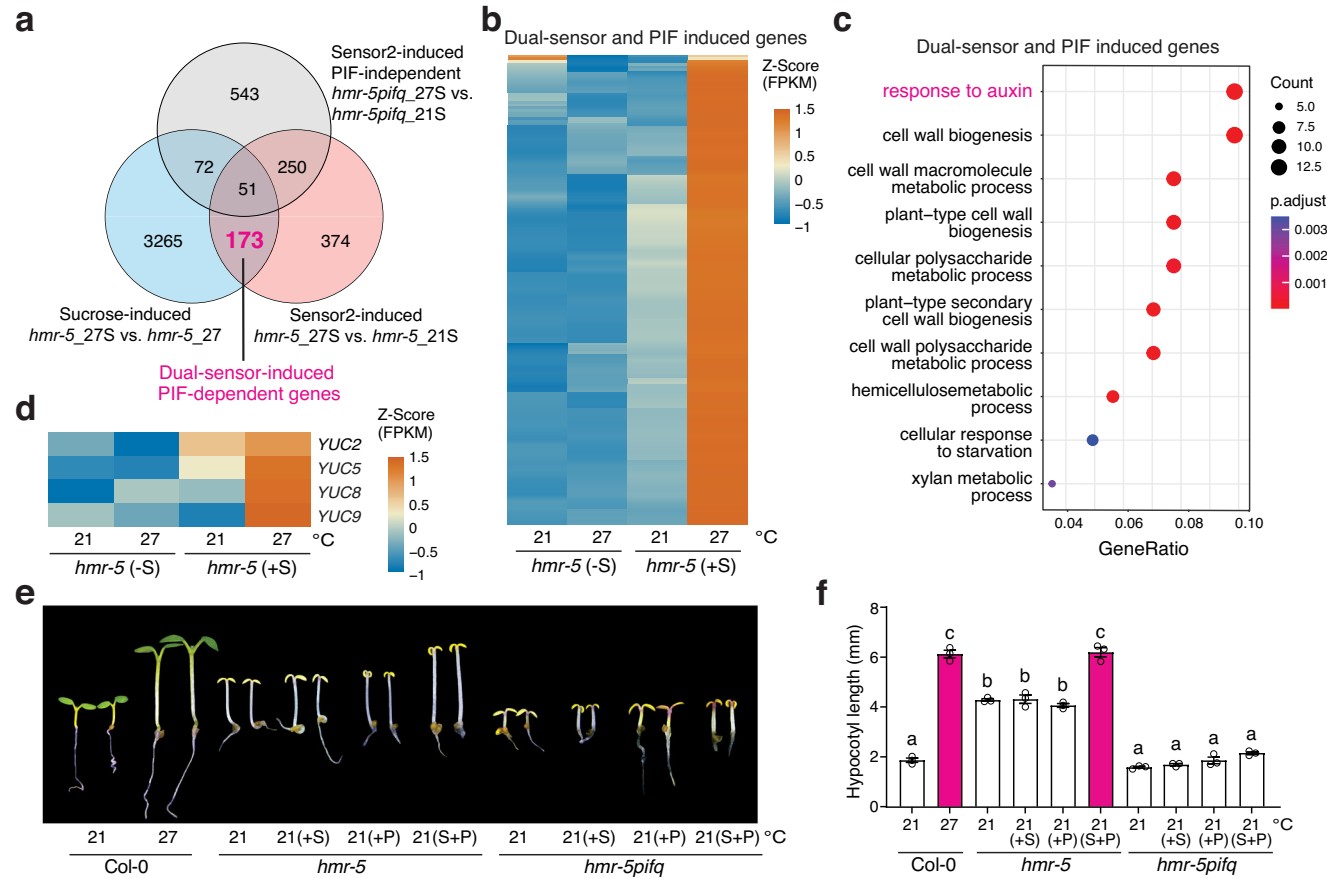

**Fig. 4 | Concerted actions of dual thermosensory pathways trigger PIF4 activity.** **a** Venn diagram depicting the 173 PIF-dependent genes whose induction requires both the chloroplast-sucrose-mediated and Sensor 2-dependent thermosensory pathways. Sucrose-induced genes: induced genes comparing *hmr-5_27S* with *hmr-5_27*; Sensor2-induced genes: induced genes comparing *hmr-5_27S* with *hmr-5_21S*; Sensor2-induced PIF-independent genes: induced genes comparing *hmr-5pifq_27S* with *hmr-5pifq_21S*. **b** Heatmap showing the relative expression levels of the 173 dual-sensor-induced and PIF-dependent genes in *hmr-5* at 21 and 27 °C with or without sucrose. **c** GO enrichment analysis of the 173 dual-sensor-induced and PIF-dependent genes. **d** Heatmap showing the relative expression levels of four

auxin biosynthesis genes, *YUC2, YUC5, YUC8*, and *YUC9*, in *hmr-5* at 21 and 27 °C with or without sucrose treatment. **e** Images of 4-day-old Col-0, *hmr-5*, and *hmr-5pifq* seedlings grown under 50 µmol m⁻² s⁻¹ R light at 21 °C on regular growth media or media with sucrose (S), picloram (P), or both sucrose and picloram (S + P). **f** Hypocotyl length measurements of the Col-0, *hmr-5*, and *hmr-5pifq* seedlings described in (**e**). Error bars represent the s.e. ($n$ = three biological replicates), and the centers of the error bars indicate the mean. Different lowercase letters denote statistically significant differences in hypocotyl length (one-way ANOVA, Tukey's HSD, $p < 0.05$, $n$ = 3 biological replicates). The underlying source data for the hypocotyl measurements in (**f**) are provided in the Source Data file.

was associated with auxin response (Fig. 4c), which included well-characterized marker genes in auxin signaling, such as *IAA19, IAA5*, and *SAUR* (Supplementary Data 1)[54–56]. These results indicate that the PIF-dependent auxin-responsive genes require not only sucrose-induced PIF4 accumulation but also the release of the antagonizing function of Sensor 2. Because PIF4 activates genes associated with both auxin biosynthesis and signaling, to test whether thermoresponsive auxin biosynthesis requires both thermosensory pathways, we examined the expression of four *YUC* genes, *YUC2, YUC5, YUC8*, and *YUC9*, that are involved in high-temperature-induced auxin biosynthesis[57,58]. In *hmr-5*, all four *YUC* genes could be induced only in the presence of sucrose and at 27 °C (Fig. 4d), suggesting that thermoresponsive auxin biosynthesis requires the concerted actions of sucrose and Sensor 2. Supporting this conclusion, supplementing the growth media with both sucrose and the synthetic auxin picloram was able to trigger hypocotyl growth in *hmr-5* at 21 °C to a similar extent as in Col-0 at 27 °C, and the same treatment failed to induce a response in *hmr-5pifq* (Fig. 4e, f). In contrast, neither sucrose nor picloram alone could promote hypocotyl growth in *hmr-5* at 21 °C (Fig. 4e, f). Thus, we concluded that the PIF4-dependent induction of auxin biosynthesis and signaling requires the concerted actions of the chloroplast-sucrose-mediated and Sensor 2-dependent thermosensory mechanisms.

## ELF3-dependent thermal regulation of PIF4 activity

The fact that sucrose alone can induce PIF4 accumulation but not the PIF4-dependent induction of auxin-associated genes implies that Sensor 2 represses PIF4 activity at 21 °C and its inhibitory function is released at 27 °C. One candidate for Sensor 2 is ELF3. ELF3 was not previously considered to be a thermosensor in LD or continuous light conditions because *elf3* mutants can still respond to high temperatures[34,35]. However, if ELF3 is involved in the Sensor 2 function, the multisensor model provides an explanation that reconciles the thermal responsiveness in *elf3*, as a mutant lacking Sensor 2 would remain sensitive to high temperatures due to the presence of the chloroplast-sucrose-mediated thermosensory pathway. It was previously shown that under continuous R light conditions, ELF3 inhibits hypocotyl growth independently of the Evening Complex by interacting directly with PIF4 to repress PIF4's transcriptional activity[59]. Therefore, we reasoned that if ELF3 is or is associated with Sensor 2, the *elf3* mutant should remain thermoresponsive in all thermo-inducible events regulated by the chloroplast-sucrose-mediated thermosensory pathway, including starch degradation, PIF4 accumulation, induction of PIF4 target genes, and hypocotyl growth. Moreover, supplementing with sucrose should stimulate both PIF4 accumulation and hypocotyl growth in *elf3*, even at 21 °C.

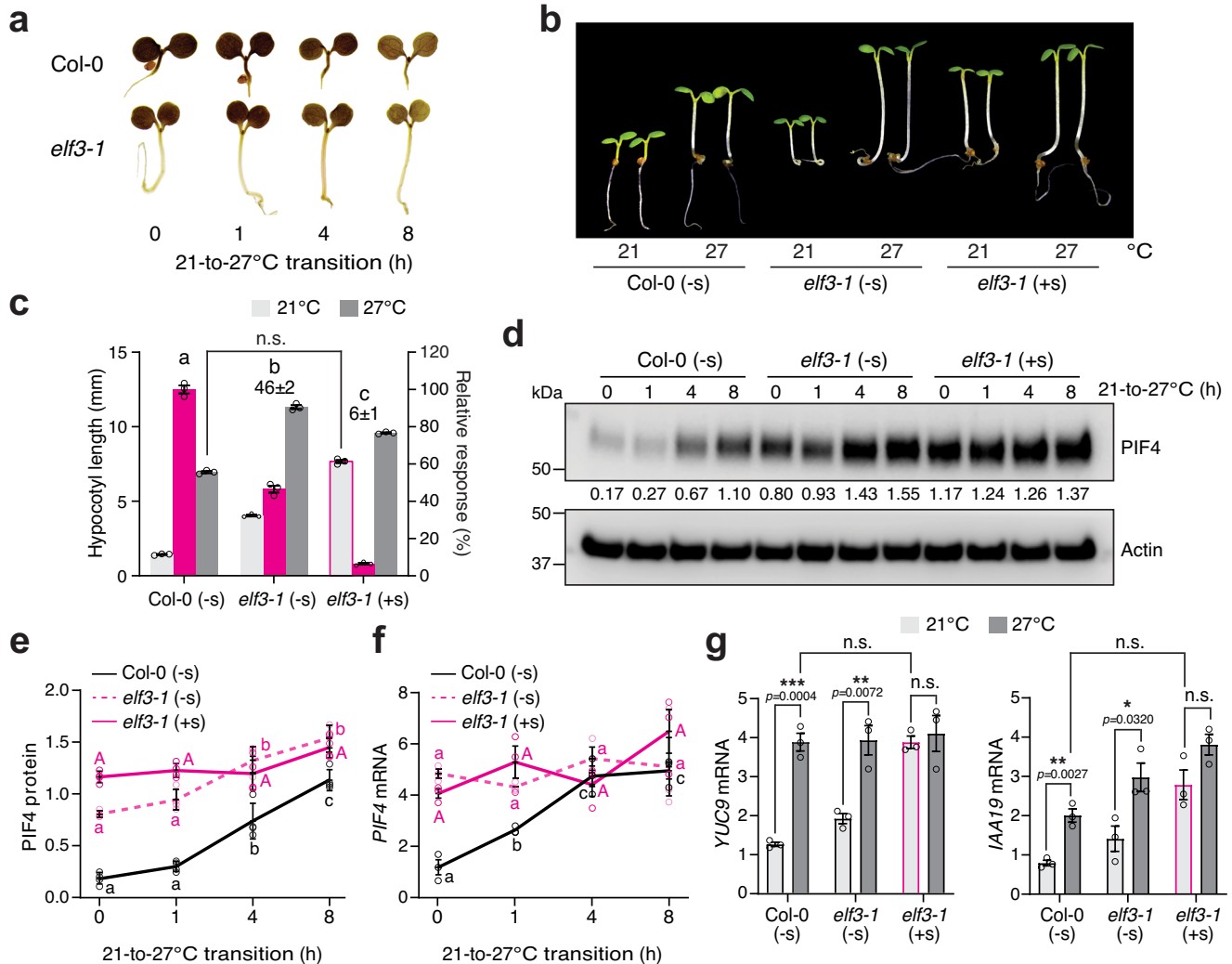

**Fig. 5 | ELF3-dependent thermal regulation of PIF4 activity. a** Iodine staining showing the starch contents of 4-d-old Col-0 and *elf3-1* seedlings grown under 50 μmol m⁻² s⁻¹ R light during the 21–27 °C transition. **b** Images of 4-d-old Col-0 and *elf3-1* seedlings grown at either 21 or 27 °C without (-s) or with (+s) sucrose. **c** Hypocotyl length measurements of the Col-0 and *elf3-1* seedlings described in (**b**). Error bars for the hypocotyl measurements represent the s.e. (*n* = 3 biological replicates), and the centers of the error bars indicate the mean. The magenta bars show the relative response, which is defined as the hypocotyl response of a mutant at 27 °C relative to that of Col-0 (set at 100%). The relative thermal responses for *elf3-1* without or with sucrose are shown. Different letters denote statistically significant differences in the relative response (one-way ANOVA, Tukey's HSD, *p* < 0.05, *n* = 3 biological replicates). *elf3-1* seedlings treated with sucrose grown at 21 °C were slightly taller than Col-0 grown at 27 °C, but the difference was within twofold and therefore considered not significant (n.s, two-tailed Student *t*-test). **d** Immunoblot analysis of the PIF4 levels in 4-d-old Col-0 and *elf3-1* seedlings grown under 50 μmol m⁻² s⁻¹ R light without or with sucrose during the 21–27 °C transition. Actin was used as a loading control. The relative PIF4 levels normalized to the corresponding levels of actin are shown. **e** Quantification of the PIF4 levels shown in

(**d**). Error bars represent the s.d. (*n* = four biological replicates), and the centers of the error bars indicate the mean. **f** qRT-PCR analysis of the *PIF4* transcript levels in 4-d-old Col-0 and *elf3-1* seedlings grown under 50 μmol m⁻² s⁻¹ R light without or with sucrose during the 21–27 °C transition. Error bars represent the s.e. (*n* = three biological replicates), and the centers of the error bars indicate the mean. For (**e** and **f**), different letters denote statistically significant differences among the values in the same genotype (one-way ANOVA, Tukey's HSD, *p* < 0.05, *n* = 3 biological replicates). **g** qRT-PCR analysis of the transcript levels of *YUC9* and *IAA19* in 4-d-old Col-0 and *elf3-1* seedlings grown under 50 μmol m⁻² s⁻¹ R light without or with sucrose at either 21 or 27 °C. Asterisks indicate a statistically significant difference based on a two-tailed Student *t*-test (* *p* < 0.05, ** *p* < 0.01, *** *p* < 0.001); n.s. stands for no significant difference. For (**f** and **g**), the transcript levels were quantified relative to those of *PP2A*. Error bars represent the s.e. (*n* = three biological replicates), and the centers of the error bars indicate the mean. The underlying source data for the hypocotyl measurements in (**c**), the immunoblots in (**d** and **e**), and the qRT-PCR analysis of transcript levels in (**f** and **g**) are provided in the Source Data file.

We next tested whether ELF3 could be Sensor 2 based on the above predictions. Indeed, thermo-inducible starch degradation remained intact in *elf3-1* compared with Col-0 (Fig. 5a). Consistent with previous reports on the photoperiod-dependent phenotypes of *elf3*[34,35], high temperatures could induce hypocotyl growth in *elf3-1* to a similar extent as in Col-0 under intense R light conditions, despite *elf3-1* being taller than Col-0 at both 21 and 27 °C (Fig. 5b, c). The PIF4 level in *elf3-1* could still be elevated by high temperatures (Fig. 5d, e). These results support the hypothesis that the chloroplast-sucrose-

mediated thermosensory pathway promotes PIF4 stability independently of ELF3. Notably, the PIF4 level was significantly higher in *elf3-1* than in Col-0 at 21 °C (Fig. 5d, e). This difference could be attributed to the elevated level of *PIF4* transcripts in *elf3-1* (Fig. 5f). Interestingly, the *PIF4* transcript accumulated to a much higher level in *elf3-1* than in Col-0 at 21 °C, and it remained at similar levels in *elf3-1* during the 21–27 °C transition, suggesting that ELF3 also plays a role in repressing *PIF4* transcription under continuous R light conditions (Fig. 5f). The fact that *PIF4* transcripts in *elf3-1* remained unchanged during

the 21–27 °C transition also suggests that the thermo-induced accumulation of PIF4 in *elf3-1* was caused mainly by sucrose-dependent PIF4 stabilization (Fig. 5e, f). Consistent with the temperature-dependent changes in PIF4 abundance, the transcripts of the PIF4 targets *YUC9* and *IAA19* were elevated in *elf3-1* by high temperatures (Fig. 5g).

Strikingly, all thermoresponsive events in *elf3-1* at both the molecular and phenotypic levels were lost with exogenous sucrose (Fig. 5b–f). Exogenous sucrose abolished the thermoresponsive accumulation of PIF4 and induction of PIF4 target genes in *elf3-1* (Fig. 5d, e, g). Unlike *hmr-5*, which requires both exogenous sucrose and auxin to trigger hypocotyl growth at 21°C (Fig. 4e, f), supplementing with sucrose alone could trigger hypocotyl growth in *elf3-1* to a similar extent as in Col-0 at 27 °C (Fig. 5b, c). With sucrose, the PIF4 level and the expression of PIF4 target genes in *elf3-1* increased even at 21 °C and remained unchanged by the elevated temperature (Fig. 5d, e, g). In contrast, the transcript level of *PIF4* stayed the same regardless of sucrose, suggesting that ELF3 regulates *PIF4* transcription independently of the chloroplast-sucrose pathway (Fig. 5f). Together, these results support the conclusion that ELF3 may act as or is associated with Sensor 2. Therefore, daytime thermo-morphogenesis is triggered by the concerted actions of chloroplast-sucrose-mediated and ELF3-dependent thermosensory mechanisms, which converge on the thermal regulator PIF4 to enable its accumulation and activity.

## Discussion

High temperatures occur concomitantly with strong sunlight during the daytime. Despite extensive studies on thermosensing in plants, the daytime thermosensory mechanism has remained obscure. In particular, it was unclear whether phyB and ELF3 could play a significant role in thermosensing during the daytime under intense light[3,4]. Here, we showed that increasing R light intensity can incrementally repress phyB thermal-reversion-dependent thermosensing, although phyB can still play a substantial role in temperature sensing and phyB's thermosensing role becomes negligible only when the R light intensity reaches 50 µmol m$^{-2}$ s$^{-1}$. Surprisingly, while temperature increases at the low-temperature range from 12 to 21 °C are sensed solely by phyB thermal reversion, temperature increases at the high-temperature range from 21 to 27 °C—which triggers thermomorphogenesis—are sensed by multiple thermosensory pathways (Fig. 6a). Leveraging a strong light condition that restricts phyB thermosensing, we unveiled the genetic circuitry of two additional thermosensory pathways required for initiating thermoresponsive hypocotyl growth in the light. High temperatures induce starch degradation in chloroplasts and subsequent production of sucrose, which stabilizes PIF4 by inhibiting phyB-dependent PIF4 degradation in the light (Fig. 6a). In parallel, high temperatures also induce *PIF4* transcription and license PIF4 activity by releasing their inhibition by ELF3 (Fig. 6a). The concerted actions of thermo-induced PIF4 stabilization by sucrose as well as the de-repression of ELF3's inhibitory functions on PIF4 transcription and PIF4 activity trigger the activation of genes associated with auxin biosynthesis and signaling to promote hypocotyl growth (Fig. 6a). This model also explains the dependency of the thermal response on phyB. In the *phyB-9* mutant, the lack of phyB-dependent PIF4 degradation and of ELF3 stabilization resulted in constitutive PIF4 accumulation at both low and high temperatures (Fig. 3a)[16,59,60], bypassing the thermal control by the chloroplast-sucrose-mediated and ELF3-mediated pathways (Fig. 6a). As such, the thermal responses can be enhanced under shade conditions, where the phyB activity is reduced[13]. For this reason, the phyB thermal reversion mechanism plays a prominent role in thermosensing at night because it dictates the kinetics of phyB inactivation and thus the abundance of active phyB in the dark[16,25]. Because PIF4 is a central thermal regulator orchestrating all aspects of thermomorphogenesis in plant

development, growth, and immunity[1,2], this study reveals a high-temperature signaling framework for understanding diverse thermo-inducible plant responses in daylight.

Our results reveal a previously uncharacterized plastid-to-nucleus retrograde signaling pathway linking thermo-inducible starch breakdown in chloroplasts to PIF4 stabilization in the nucleus. We demonstrate that a temperature increase from 21 to 27 °C induces rapid starch degradation in chloroplasts and sucrose production in the light. The diurnal regulation of starch biosynthesis and degradation has been extensively studied. In many plants, including Arabidopsis, a substantial proportion of fixed carbon produced via photosynthesis is stored in starch in the daytime and remobilized to support maintenance and growth at night[61]. We reveal that elevated temperatures can unexpectedly induce starch degradation under strong light conditions (Fig. 2e, f). The mechanism by which high temperatures trigger starch degradation remains elusive, though similar to dark- and cold-induced starch degradation, thermo-inducible starch breakdown also relies on SEX1-mediated starch phosphorylation[44,47,49,51]. Although it is tempting to propose that chloroplasts may be the thermosensor that triggers starch degradation, the current data cannot rule out the possibility that starch degradation in the chloroplasts is triggered by a thermosensor localized outside the chloroplasts. The essential role of starch breakdown in thermomorphogenesis had not been demonstrated previously. Mutants attenuating either starch degradation (*sex1-8*) or sucrose signaling (*tps1-12*) were impaired in thermo-induced PIF4 accumulation, PIF4-dependent activation of auxin-relevant genes, and hypocotyl growth (Figs. 2c, d and 3d–g). These results are consistent with a previous study showing that sucrose signaling via Tre6P is required for PIF4 stabilization and thermomorphogenesis[53]. Utilizing the chloroplast-deficient *hmr* mutants allowed us to dissect the specific function of chloroplasts and sucrose in antagonizing phyB-mediated PIF4 degradation. In *hmr-5* and *hmr-22*, where the chloroplast-dependent thermal-inducible sucrose production pathway is abolished, thermoresponsive hypocotyl growth is impeded because PIF4 accumulation remains inhibited by phyB at high temperatures (Figs. 2c, d, 3a, b, and 6b)[16,25]. Strikingly, supplementing with sucrose can promote PIF4 accumulation at both 21 and 27 °C and restore the thermal response in *hmr* mutants (Figs. 2c, d, 3a, b, and 6c), lending further support to the essential role of chloroplasts and sucrose in PIF4 stabilization and thermoresponsive hypocotyl growth (Fig. 6b, c). Although sucrose can promote growth as an energy source and a signal, it is the signaling role of sucrose that triggers thermoresponsive hypocotyl growth. This conclusion is supported by the following lines of evidence. First, the sucrose signaling mutant *tps1-12* lacks the thermal response despite normal chloroplast development (Fig. 2c, d). While sucrose supplementation restores the thermo-inducible hypocotyl response in *sex1-8*, it did not restore the temperature response in *tps1-12*, indicating that Tre6P signaling mediates thermoresponsive hypocotyl growth downstream of sucrose production (Fig. 2c, d). Second, the hypocotyl growth defect in *hmr-5* is phyB-dependent and not due to the lack of energy, because *hmr-2/phyB-9* was almost as tall as *phyB-9*[43]. Notably, blocking the chloroplast-sucrose-mediated pathway in *hmr*, *sex1-8*, and *tps1-12* did not affect the thermo-inducible accumulation of *PIF4* transcripts (Fig. 3e, h)[16]. Therefore, the chloroplast-sucrose-mediated thermosensory pathway acts specifically on PIF4 stabilization, not on *PIF4* transcription (Fig. 6a–c). Tre6P inhibits the kinase complex SUCROSE-NON-FERMENTING-1-RELATED PROTEIN KINASE (SnRK1)—an evolutionarily conserved control point for energy signaling in eukaryotes[62–64]. Tre6P has been proposed to stabilize PIF4 by attenuating SnRK1-mediated PIF4 phosphorylation[53]. Changes in sucrose allocation from the cotyledons to the hypocotyl have also been shown to play an important role in shade-induced hypocotyl growth[65]. Thus, our results may suggest that sucrose transportation and signaling play similar roles in both temperature- and light-induced hypocotyl growth.

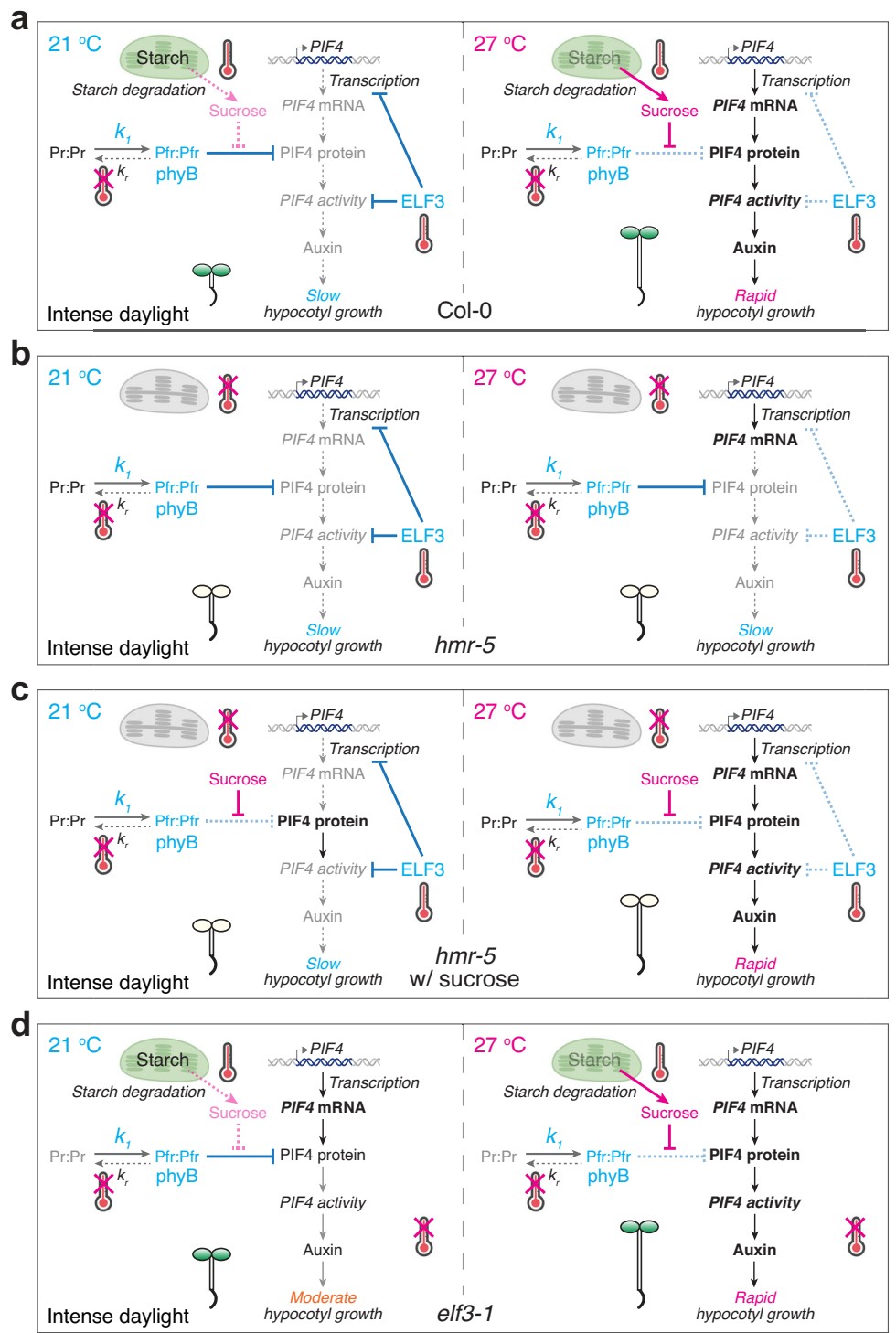

We have previously shown that HMR and REGULATOR OF CHLOROPLAST BIOGENESIS (RCB) are dual-targeted proteins localizing to chloroplasts and the nucleus and are required for phyB-mediated PIF1 and PIF3 degradation and the formation of phyB-containing subnuclear photobodies[16,25,43,66,67]. It was puzzling that while HMR and RCB are required for PIF1 and PIF3 degradation, they are essential for PIF4 stabilization[16,25]. Our new results indicate that the primary cause for the defect in PIF4 accumulation in the *hmr* mutants is the lack of chloroplasts and photosynthesis, as supplementing with sucrose was sufficient to restore PIF4 accumulation and thermo-responsive hypocotyl growth in *hmr-5* and *hmr-22* (Figs. 2c, d and 3a, b). This may explain the discovery of *rcb* mutants as suppressors of

*hmr-22* as *rcb-101/hmr-22* rescues both the greening and thermomorphogenesis defects of *hmr-22*[25]. Supplementing with sucrose did not rescue the tall hypocotyl phenotype of the *hmr* mutants at 21 °C (Fig. 2c, d), as the tall hypocotyl phenotype of *hmr* depends on the accumulation of PIF1 and PIF3, implying that HMR's role in regulating PIF1 and PIF3 degradation is separate from sucrose signaling[25,43,66–68].

Our results unveil an ELF3-dependent daytime thermosensory pathway in which high temperatures increase *PIF4* transcripts and license PIF4 activity by releasing their inhibition by ELF3 (Fig. 6a). It has been previously shown that ELF3 can interact directly with PIF4 and inhibit its DNA-binding activity, independently of the Evening Complex, to regulate hypocotyl growth in the light[59]; this mode of action is

**Fig. 6 | A multisensor high-temperature signaling framework for triggering daytime thermomorphogenesis.** Three thermosensory mechanisms – phyB thermal-reversion-dependent, chloroplast-sucrose-mediated, and ELF3-dependent −work collaboratively and converge on the central thermal regulator PIF4 to trigger thermomorphogenesis in the light. **a** At 21 °C, phyB represses PIF4 accumulation by promoting its degradation, and ELF3 inhibits both *PIF4* transcription and PIF4 activity; consequently, PIF4 is repressed at the transcript, protein, and activity levels. In contrast, at 27 °C, thermo-induced starch degradation in the chloroplasts and sucrose production promote PIF4 accumulation by inhibiting phyB-mediated PIF4 degradation. In parallel, the ELF3-dependent inhibition of *PIF4* transcription and PIF4 activity is released. The combined induction of PIF4 accumulation and activity triggers the transcriptional activation of growth-relevant PIF4 target genes, including those associated with auxin biosynthesis and signaling, thereby triggering thermomorphogenesis responses such as hypocotyl growth. The accelerated thermal reversion of phyB at high temperatures can also enhance PIF4 stabilization in relatively low light intensities. However, under strong light conditions where the intensity of R light reaches 50 μmol m$^{-2}$ s$^{-1}$ or above, because the rate of phyB photoactivation ($k_1$) is significantly higher than that of phyB thermal reversion ($k_r$), the effect of phyB thermal-reversion-dependent thermosensing becomes negligible, leaving only the chloroplast-sucrose-mediated and ELF3-dependent thermosensing mechanisms operational, as shown in the model. **b** In *hmr-5*, under intense light conditions, both the phyB-dependent and chloroplast-sucrose-mediated thermosensory mechanisms are eliminated, and thermomorphogenesis is regulated by the ELF3-dependent thermosensing mechanism only. In this scenario, although the level of *PIF4* transcripts can still be enhanced by high temperatures, PIF4-mediated thermomorphogenesis is blocked because PIF4 accumulation is repressed by phyB at both temperatures. **c** In *hmr-5* supplemented with sucrose, exogenous sucrose promotes PIF4 accumulation by antagonizing phyB-mediated PIF4 degradation, thereby bypassing the defect in the chloroplast-sucrose-mediated thermosensory mechanism. However, although the levels of PIF4 can be enhanced at both temperatures, sucrose alone cannot trigger thermomorphogenesis at 21 °C because PIF4 activity is still repressed by ELF3. Therefore, in this scenario, thermomorphogenesis is regulated solely by the ELF3-dependent thermosensory mechanism. **d** In *elf3-1*, under intense light conditions, only the chloroplast-sucrose-mediated thermosensory mechanism remains operational. As a result, exogenous sucrose can fully turn on thermomorphogenesis at 21 °C. The image elements of chloroplasts, thermometers, and double-stranded DNA were created in BioRender. Chen, M. (2025) https://BioRender.com/9wqzqu2.

distinct from the thermosensing function of ELF3 in repressing *PIF4* transcription through the Evening Complex at night[20,26,30]. Mathematical modeling has predicted that photothermal-switchable hypocotyl elongation under a combination of high-temperature and intense R light conditions requires an unknown factor, Y, which promotes PIF4 activity at high temperatures[69], and ELF3 was proposed to be Y[59]. Our results indicate that high temperatures license PIF4 transcriptional activity by releasing its inhibition by ELF3 (Figs. 5 and 6a). In the *elf3-1* mutant grown under strong light, where the ELF3-dependent thermosensory pathway is turned on constitutively, thermosensing would depend solely on the chloroplast-sucrose-mediated pathway (Fig. 6d). Consistent with this prediction, exogenous sucrose alone could stimulate hypocotyl growth in *elf3-1* at 21 °C to the same extent as in Col-0 at 27 °C, and all the phenotypic and molecular thermal responses of *elf3-1*, including the expression of PIF4 target genes, became thermo-insensitive in the presence of sucrose (Fig. 5). In contrast, in *hmr-5*, the expression of auxin biosynthesis and signaling genes and triggering thermoresponsive hypocotyl growth require both exogenous sucrose and the release of the inhibition of PIF4 activity by ELF3 (Figs. 4 and 6c). It was previously shown that ELF3 can directly sense high-temperature via thermo-inducible ELF3 condensation[20] and that ELF3 is a component of phyB-containing photobodies[70,71]. ELF3 may also act as a thermosensor during the daytime via a similar mechanism. For example, it is possible that high temperatures could induce the sequestration of ELF3 to photobodies or other condensates, thereby releasing ELF3-dependent inhibition of PIF4 activity and *PIF4* transcription in the nucleoplasm. However, the current data cannot exclude the possibility that the ELF3-dependent daytime thermosensory pathway is initiated by another thermosensor. For example, THERMO-WITH ABA-RESPONSE 1 (TWA1) senses high temperatures to induce the transcription of *HEAT SHOCK TRANSCRIPTION FACTOR A2* (*HSFA2*) and heat shock proteins[72]. Some HSFAs, such as HSFA1d and its paralogs, play essential roles in stabilizing PIF4 during the daytime by disrupting the phyB-PIF4 interaction[73]. HSFA1 also triggers the heat-induced depletion of repressive H2A.Z-nucleosomes and the expression of target genes[74]. Interestingly, the expression of *HSFA1* and HSFA1-associated thermotolerance are gated by light-dependent chloroplast stresses, including perturbations in starch synthesis[75]. Further studies are needed to elucidate the role of the TWA1- and HSFA-mediated thermotolerance mechanisms in thermomorphogenesis. Moreover, because the *PIF7*-transcript-dependent RNA thermoswitch also operates during the daytime[14] and PIF7 may heterodimerize with PIF4, the *PIF7* pathway could also contribute to the regulation of PIF4 activity by ELF3.

In conclusion, our study elucidates a multisensor-mediated high-temperature signaling framework for triggering daytime thermomorphogenesis in Arabidopsis. The chloroplast-sucrose-mediated and ELF3-dependent daytime thermosensory pathways converge on the nodal thermal regulator PIF4. Because activation of these two thermosensing pathways individually is insufficient to induce the full hypocotyl growth response, these two thermosensory mechanisms represent a double-input AND logic gate. One advantage of the "AND gate" design could be to improve the thermosensing specificity; it would allow each thermosensory pathway to operate individually without inducing a growth output. Future studies will focus on testing these hypotheses and elucidating the thermosensing mechanisms.

## Methods

### Plant materials, growth conditions, and hypocotyl measurements

The Columbia (Col-0) and Landsberg *erecta* (L*er*) ecotypes of *Arabidopsis thaliana* were used throughout this study. The Arabidopsis mutants *phyB-9*[76], *elf3-1*[29], *pif4-2* (SAIL_1288_E07)[9], *pifq*[9], *hmr-5*[68], *hmr-22*[66], *hmr-5pifq*[66], *tps1-12* (CS2109724)[77], and *sex1-8* (SALK_077211C)[50] were previously described. The *YHB* line, which expresses phyB$^{Y276H}$ under the 35S promoter in *phyB-9*, was previously described[41,42,78]. The transgenic *YHB*$^g$ line, which expresses a genomic copy of *PHYB*$^{Y276H}$ under the native *PHYB* promoter in *phyB-5* (L*er*), was also previously described[37,79].

Arabidopsis seeds were surface-sterilized and plated on half-strength Murashige and Skoog media with Gamborg's vitamins (MSP06, Caisson Laboratories, Smithfield, UT), 0.5 mM MES pH 5.7, and 0.8% (w/v) Phytoagar (40100072-2, PlantMedia, Dublin, OH). For sucrose treatments, the growth media was supplemented with 1% (w/v) sucrose (8550-5KG, MilliporeSigma, Burlington, MA). For picloram treatments, the growth media was supplemented with 5 μM picloram (P5575-10G, Sigma-Aldrich, St. Louis, MO). Seeds were stratified in the dark at 4 °C for 5 days before treatment under specific light and temperature conditions in an LED chamber (Percival Scientific, Perry, IA). Fluence rates of light were measured using an Apogee PS200 spectroradiometer (Apogee Instruments, Logan, UT) and SpectraWiz spectroscopy software (StellarNet, Tampa, FL). For the dark-grown samples, seeds were exposed to 10 μmol m$^{-2}$ s$^{-1}$ FR light for 3 h after stratification to induce germination. For temperature treatment, seedlings were initially grown under 50 μmol m$^{-2}$ s$^{-1}$ R light at 21 °C for 2 days. Subsequently, they were either maintained at 21 °C or transferred to 27 °C under the same light condition for two additional days. For the 21–27 °C transition experiments, seedlings were grown

under 50 µmol m$^{-2}$ s$^{-1}$ R light at 21 °C for 4 days and then transferred to 27 °C under the same light condition.

The method for hypocotyl measurements has been previously described[34]. At least 60 seedlings from each genotype were scanned using an Epson Perfection V700 photo scanner (Epson America, Los Alamitos, CA), and hypocotyl length was measured using NIH ImageJ software (http://rsb.info.nih.gov/nih-image/). Bar charts were generated using GraphPad Prism 8 (GraphPad Software, San Diego, CA). Images of representative seedlings were taken using a Leica M165 FC microscope (Leica Microsystems Inc., Buffalo Grove, IL) and processed using Adobe Photoshop v22.3.0 (Adobe Inc., Mountain View, CA).

### Protein extraction and immunoblotting

Total protein was extracted from Arabidopsis seedlings grown under the indicated conditions. A total of 100 mg plant tissue was ground in liquid nitrogen and resuspended in extraction buffer containing 100 mM Tris-HCl pH 7.5, 100 mM NaCl, 5% SDS, 5 mM EDTA pH 8.0, 20 mM dithiothreitol, 20% glycerol, 142 mM β-mercaptoethanol, 2 mM phenylmethylsulfonyl fluoride, 1× cOmplete™ EDTA-free Protease Inhibitor Cocktail (SIAL-11836170001, Sigma-Aldrich), 80 µM MG132, 80 µM MG115, 1% phosphatase inhibitor cocktail 3 (P0044, Sigma-Aldrich), 10 mM N-ethylmaleimide, 2 mM sodium orthovanadate (Na$_3$OV$_4$), 25 mM β-glycerophosphate disodium salt hydrate, 10 mM NaF, and 0.01% bromophenol blue. Protein extracts were boiled for 10 min and then centrifuged at 16,000 × $g$ for 10 min at room temperature. Protein extracts were separated via 10% SDS-PAGE, transferred to a PVDF membrane, probed with the indicated primary antibodies, and then incubated with horseradish peroxidase (HRP)-conjugated secondary antibodies. Rabbit anti-PIF4 polyclonal antibody (R2534-4, Abiocode, Agoura Hills, CA) was used at a 1:1000 dilution. Mouse anti-phyB monoclonal antibody (a gift from Drs. Akira Nagatani and Yoshito Oka) and monoclonal mouse anti-actin antibodies (A0480, Sigma-Aldrich) were used at a 1:2000 dilution. HRP-conjugated goat anti-mouse (1706516, Bio-Rad Laboratories, Hercules, CA) and goat anti-rabbit (1706515, Bio-Rad Laboratories) secondary antibodies were used at a 1:5000 dilution. Signals were detected via SuperSignal West Dura Extended Duration Chemiluminescent Substrate (PI34076, Thermo Fisher Scientific, Waltham, MA). Images of immunoblots were taken using a LI-COR Odyssey XF (LI-COR, Lincoln, NE). Western blots were quantified using LI-COR Image Studio software (LI-COR).

### RNA extraction and quantitative real-time PCR

A total of 100 mg of seedlings was collected and flash-frozen in liquid nitrogen. Plant tissues were ground to a fine powder in liquid nitrogen. Total RNA was extracted using a Quick-RNA MiniPrep kit with on-column DNase I digestion (R1055, Zymo Research, Irvine, CA). cDNA synthesis was performed with 1 µg total RNA using SuperScript II Reverse Transcriptase (18064014, Thermo Fisher Scientific, Waltham, MA), and oligo(dT) primers (18418012, Thermo Fisher Scientific) were used for all target genes. qRT-PCR was performed with iQ SYBR Green Supermix (1708880, Bio-Rad Laboratories) on a Roche LightCycler 96 system (Roche, Basel, Switzerland). The mRNA level of each gene was normalized to that of *PP2A* (At1G13320). The primers for qRT-PCR analysis used in this study are listed in Supplementary Table 1.

### RNA-seq and data analysis

RNA sequencing was performed by the Novogene Corporation Inc. (Sacramento, CA). Ribosomal RNA was removed from the total RNA, followed by ethanol precipitation. After fragmentation, first-strand cDNA was synthesized using random hexamers. During second-strand cDNA synthesis, dUTPs were replaced with dTTPs in the reaction buffer. The directional library was ready after end repair, A-tailing, adapter ligation, size selection, enzyme digestion to remove UTP-containing second-strand cDNA, amplification, and purification. The libraries were subjected to paired-end 150 bp sequencing using a NovaSeq 6000 (Ilumina, Inc., San Diego, CA). RNA-seq data analysis was performed using the pRNASeqTools pipeline (https://github.com/grubbybio/pRNASeqTools). Briefly, the raw reads were mapped to the Araport11 genome using STAR version 2.7.7a, and the number of reads mapped uniquely to each annotated gene was counted using FeatureCounts version 2.0.1. Transcript levels were measured in fragments per kilobase per million total mapped fragments (FPKM). Differentially expressed genes were identified using DEseq2 v1.35.0 with a fold change of two and $p < 0.01$. Heatmaps were generated using pheatmap v1.0.1278. Venn diagrams were generated using ggvenn v0.1.980. GO cluster analysis was performed using clusterProfiler 4.058.

### Starch staining

At least 30 seedlings were harvested and immersed in 3 mL of 80% ethanol at 60 °C for 5 min, followed by replacement with fresh ethanol and an additional 5-min heating period until the leaf pigments were removed. After ethanol removal, the seedlings were washed twice with deionized water for 2 min per wash and then stained in Lugol's solution (1092611000, MilliporeSigma) for 15 min at room temperature, rinsed briefly in deionized water and then photographed. Images of representative seedlings were taken using a Leica M165 FC microscope (Leica Microsystems).

### Soluble sugar and starch quantification

Starch and sucrose were extracted from approximately 200 mg of 4-day-old seedlings. After grinding in liquid nitrogen, the samples were extracted three times with methanol:chloroform:water (MCW) (12:5:3 by volume)−first with 1.5 mL and then twice with 0.5 mL. Supernatants containing soluble sugars were pooled and mixed with 0.6 volumes (1.5 mL) of water and centrifuged at 16,000 × $g$ for 10 min. The upper aqueous phase was collected and evaporated at 30 °C with a Vacufuge Plus (Eppendorf, Hamburg, Germany). Soluble sugars were resuspended in 500 µL of water and quantified using a Sucrose/Fructose/D-Glucose Assay Kit (Megazyme, Bray, Ireland). Starch content was measured from the pellet obtained after MCW washing using a Total Starch HK Assay Kit (Megazyme).

### Reporting summary

Further information on research design is available in the Nature Portfolio Reporting Summary linked to this article.

## Data availability

Arabidopsis mutants used in the current study are available from the corresponding authors upon request. The RNA-sequencing data associated with this study have been deposited in the GEO database with the accession code GSE275012. Source data are provided with this paper.

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

## Acknowledgements

We thank Akira Nagatani and Yoshito Oka (Kyoto University) for providing the anti-phyB antibodies. We thank Elise Pasoreck, Nora Flynn, Juan Du, and Jiangman He for their valuable suggestions and comments on the manuscript. We thank Chenjiang You (South China Agricultural University) for providing the pRNASeqTools pipeline. We thank the High-Performance Computing Center at the University of California, Riverside, for providing the computational resources necessary for the bioinformatics tools used in the study. This study was supported by National Institutes of Health grants R01GM087388 and R01GM132765 to M.C. and 5R35GM139598 to J.C.L., National Science Foundation grant MCB-2141560 to M.C., USDA-NIFA Hatch project CA-R-BPS-5186-H to M.C., and a Howard Hughes Medical Institute bridge fund to X.C.

## Author contributions

D.F., X.C., and M.C. conceived of the original research plan; M.C. and X.C. supervised the experiments; D.F., N.X., and E.R.S. performed the experiments; W.H. and J.C.L. provided the *YHB^g* line; D.F., W.H., N.X., E.R.S., J.C.L., X.C., and M.C. analyzed the data; D.F., X.C., and M.C. wrote the article with contributions from all authors.

## Competing interests

The authors declare no competing interests.
