## [Peer Review file · Nature Communications]

A multisensor high-temperature signaling framework for triggering daytime thermomorphogenesis in Arabidopsis

Corresponding Author: Professor Meng Chen

Version 0:

Reviewer comments:

Reviewer #1

(Remarks to the Author)

The authors examined hypocotyl elongation by varying the red fluence rate and temperature. They concluded that the temperature response could be divided into a phyB-dependent response (12–21°C) and a phyB-independent response (21–27°C). They showed that the phyB-independent temperature response (hypocotyl elongation) is induced by temperature-triggered starch degradation in chloroplasts. This conclusion is supported by the loss of the temperature response in *sex1* and *tps1* mutants and its rescue with sucrose supplementation. The *hmr* mutant, which disrupts both nuclear phyB signaling and chloroplast development, also behaves similarly to the *sex1* and *tps1* mutants. Curiously, however, the authors observed that the *hmr* mutant supplemented with sucrose at 21°C accumulates PIF4 protein at levels comparable to those at 27°C. Yet, the hypocotyl length of the mutant is shorter at 21°C and elongates further when the temperature increases to 27°C. This suggests the presence of another thermosensor that inhibits PIF4 activity. The authors proposed that ELF3 acts as this additional thermosensor. The manuscript provides valuable insights into the complex nature of plant thermoresponses. Below are my comments:

Temperature Response Classification:

I agree that temperature responses inducing hypocotyl elongation can have multiple facets depending on the temperature. However, dividing the temperature response into "phyB-dependent" and "phyB-independent" categories at 21°C is misleading. The thermal reversion rate of phyB is theoretically proportional to temperature across a broad range, and 21°C is unlikely to serve as a distinct dividing point. It is more plausible that phyB-dependent responses operate throughout the temperature range used in the experiments. The observed "phyB-independent" responses at 21–27°C might be due to the increased dominance of phyB-independent processes at higher temperatures. Terms like "phyB-dominant" and "non-phyB-dominant (?)" might be more appropriate. Additionally, the lack of data points between 21°C and 27°C could contribute to the perception of 21°C as an arbitrary dividing temperature.

Experimental Verification using *tps* mutant:

Since the authors propose that one process is mediated by trehalose-6-phosphate (T6P), the separation of "phyB-dependent" and "phyB-independent" responses across the temperature range could be experimentally clarified by analyzing the fluence- and temperature-dependent hypocotyl lengths of *sex1* or *tps1* mutants. These mutants are more appropriate than the *hmr* mutant, as HMR also affects phyB nuclear signaling.

Inclusion of phyB and phy Multiple Mutant Data:

The authors' conclusions primarily rely on the temperature responses of wild-type and YHBg lines rather than phyB mutant or phy multiple mutant phenotypes. Since the authors already have access to phy mutants, incorporating this data would strengthen the study. Although interpreting such data could be complex due to the convergence of signaling pathways on PIF4, it would provide additional support for the proposed mechanisms.

Clarification of PhyB Dependency:

The authors noted that sucrose does not increase PIF4 protein levels in the phyB mutant. Does this imply that the "phyB-independent" process is actually phyB-dependent? This could be due to saturation effects. However, without definitive evidence, the authors should be more cautious in drawing conclusions about what is truly phyB-dependent or independent.

Identities of three thermosensing Systems:

While the authors suggest the presence of three independent thermosensing systems—phyB, starch degradation leading to the T6P, and ELF3—the independence of these systems are not experimentally clear. Authors should experimentally clarify the independence/dependence or should be cautious for their descriptions.

Reviewer #2

(Remarks to the Author)
Report – NCOMMS-24-85051-T

Key results:

In this manuscript, the authors investigate the molecular mechanisms that gate thermoresponsive hypocotyl elongation, a hallmark of thermomorphogenesis, during daytime, i.e., during the light period, in *Arabidopsis*. The authors find that warm temperatures induce the remobilisation of transitory starch in the light and that starch degradation is required for thermoresponsive hypocotyl elongation. Furthermore, they show that sucrose supplementation induces the accumulation of PIF4 protein, the central positive transcriptional regulator of thermomorphogenesis, but is not sufficient to induce elongation growth, thereby suggesting the existence of (at least) one additional thermosensor. By comparing the transcriptional responses to sucrose supplementation and to a temperature shift from 21 °C to 27 °C and subsequent mutant analyses the authors suggest that ELF3 acts as the second thermosensor in the light.

Strengths:

The authors nicely dissect temperature-responsive hypocotyl elongation into a phyB-dependent and phyB-independent component using thermal and light fluence gradients, thereby demonstrating the limited contribution of phyB to thermal sensing in the light. The requirement of transitory starch remobilisation in the light for warm temperature-induced hypocotyl elongation shown here suggests a very interesting link between carbon/energy metabolism and temperature-responsive growth that has been studied little in the past, providing several interesting implications for future research. In that sense, the manuscript is of interest to people within the thermomorphogenesis community. However, in my opinion several open questions remain. Also, some of the conclusions drawn based on the data are too strong and not justified as is (see below).

Weaknesses:

(1) The authors repeatedly suggest that chloroplasts act as (or harbour) a thermosensor (e.g., “chloroplasts, as essential thermosensory organelles” (line 338), or “[...] that chloroplasts play an essential role in thermosensing in daylight” (line 186)). However, this is not demonstrated in the manuscript. The SEX1-mediated starch degradation in the light at 27 °C that coincides with an increase in endogenous sucrose could also rely on temperature sensing outside of the chloroplasts. Also, the statement in the caption of Fig. 2 “Thermo-inducible starch breakdown in chloroplasts is essential for thermosensing in the light” is too bold in my opinion. I would rather carefully phrase that thermo-inducible starch breakdown is a prerequisite or required for warm temperature-induced elongation growth in the light”.

(2) The authors suggest that “high temperatures induce starch degradation in chloroplasts and subsequent sucrose production and signaling to stabilize PIF4 in daylight” (lines 299ff). However, the experiments presented do not connect starch degradation to PIF4 protein levels directly. Instead, the authors exogenously provided sucrose to different mutant lines and observed increased PIF4 protein levels. Given that the authors demonstrate that warm temperature-induced starch degradation in Col-0 coincides with enhanced endogenous sucrose levels, and that this is absent in *sex1-8* mutants, I would suggest monitoring PIF4 protein levels in this mutant as a function of temperature in parallel. According to their hypothesis, one would expect reduced PIF4 protein levels in *sex1-8* compared to the wild type. This could also be accompanied by expression analyses of well-characterised PIF4 target genes, e.g., YUCCA9, IAA19, and would finally establish a link between starch degradation and PIF4 stabilisation.

(3) Still referring to the warm temperature-induced starch degradation: are there any hypotheses/indications how SEX1-mediated starch degradation at 27 °C is induced? Does this rely on transcriptional regulation of SEX1, or rather post-translational activation? Is there anything known about the relevance of other starch degrading enzymes at elevated temperatures? I would appreciate a few sentences on this topic in the discussion.

(4) In their final model, the authors indicate that the sucrose-dependent effects on PIF4 stability are mediated via T6P (Fig. 5). However, this is not necessarily the case. While it has been shown that elevated T6P levels do indeed enhance PIF4 protein stability in a KIN10-dependent manner (Hwang et al., 2019, EMBO Reports), the effects exerted by sucrose supplementation shown in the manuscript could also be T6P-independent.

In addition, under the conditions used in their experiments (constitutive monochromatic R light), SnRK1 as a regulator of the starvation response, likely is inactive anyway. T6P-dependent stabilisation of PIF4 via repression of SnRK1, however, would require active SnRK1 in the first place to be functional. Therefore, this mechanism does not seem very likely to play a role here, in my opinion. Maybe the authors could discuss this in some more detail. As an additional suggestion, T6P-dependent, GRIK1/SnRK1-mediated effects on PIF4 protein stability (as proposed by Hwang et al., 2019, EMBO Reports) could be indicated as a potential mechanism parallel to T6P-independent effects exerted by increased sucrose levels in the final model (Fig. 5)?

(5) Concerning the experiments involving the *hmr* mutants: I was wondering whether the absence of warm temperature-responsive growth in the *hmr* mutants (in the absence of exogenous sucrose) could be a result of these seedlings being starving given that genes required for carbon assimilation and sugar production are deregulated in these mutants. Once you provide them with fuel (in the form of exogenous sucrose in this case), they can elongate in response to higher temperature. In this case, I would then argue that in these mutants the starvation response might be negatively impacting on growth thereby preventing warm temperature-responsive hypocotyl elongation. This would be different from the suggested conclusion (lines 166ff) that sucrose production may be required for thermosensing.

Minor Points:

- Lines 136ff.: the authors mention that the phyBY276H variant lacks thermal reversion. I wonder if the lack of thermal reversion has really been demonstrated for phyBY276H (because it is not shown in the two publications cited in line 137), or whether this is extrapolated from the fact that this mutant lacks photoconversion?
- Caption Fig. 1b: "Distinct light-intensity effects on phyB-dependent and -independent temperature responses. PhyB-dependent low-temperature response (12-21 °C under dim 2.5 $\mu\text{mol m}^{-2} \text{s}^{-1}$ R light) and phyB-independent high-temperature response (21-27 °C under strong 50 $\mu\text{mol m}^{-2} \text{s}^{-1}$ R light) were quantified as the $d(\text{hypocotyl length})/dT$ derivatives and plotted against light intensity." What do the underlined specifications mean in this case? The derivatives for the low temperature response and the high temperature response were plotted against the different light intensities. Therefore, specifying specific intensities in the parentheses does not make sense to me.
- Caption Fig. 3e: there is the word "genes" missing between "PIF-induced" and "in hmr-5".

The statistical tests applied appear appropriate to me, however, the significance threshold being applied appears rather arbitrary in some instances (e.g., only reporting p values smaller than 0.01 as statistically significant (Fig. 1c and e) is rather uncommon). Is there any particular reason for applying this threshold when doing a single comparison as in Fig. 1c and e? The figures are of high quality and the data is presented well. Also, the manuscript is clearly written with enough detail in the methods section.

Version 1:

Reviewer comments:

Reviewer #1

(Remarks to the Author)

All my concerns have been addressed in the revised manuscript. The addition of Figure 3 and the updated schematic diagram (Figure 6) present the co-action of multisensors in high-temperature signaling more convincingly.

Reviewer #2

(Remarks to the Author)

Report – NCOMMS-24-85051-A - Revised Manuscript

As stated before, I find the presented study very interesting and agree with reviewer #1 that it provides valuable insights into plant temperature responses. The authors addressed all my concerns, and the requested experimental data has been added. However, I do have a few minor comments.

The requested data on PIF4 accumulation and downstream target gene expression in *sex1-8* and *tps1-12* have been added and are in line with the hypothesised mechanism. I also appreciate that question whether sucrose serves as energy source or whether it might elicit a kind of "signal" has been discussed in more detail (referring to lines 404-409). Concerning this discussion, I would even add another argument in favour of the signal hypothesis: the fact that sucrose supplementation restores WT-like hypocotyl elongation in *sex1-8* but not in *tps1-12* also points towards the signalling role as crucial for thermoresponsive hypocotyl growth. With hindsight, and with regard to the question as to whether TPS1-dependent T6P signalling is upstream or downstream of starch degradation and sucrose production, it would have been interesting to assess to what extent sucrose supplementation of *sex1-8* and *tps1-12* can restore warm temperature-dependent PIF4 accumulation (referring to Fig. 3c-g) and induction of PIF4 target genes. If the sugar signal was key here, then supplementing *tps1-12* with exogenous sucrose should not be sufficient to restore PIF4 accumulation since the sugar signal is missing, while in *sex1-8* it should be sufficient. But this might be a topic for future investigations.

Concerning the presentation of the hypocotyl elongation data, I preferred the previous presentation as " $d(\text{hypocotyl length})/dT$ " over the "Relative response (%)". (If this change was done in response to one of my previous comments, I must apologise for this misunderstanding. But this was not what I meant. My previous comment just referred to the figure caption that has been modified accordingly.) Setting the WT response to 100% and calculating the effect sizes of the temperature treatment in the mutant lines relative to WT seems arbitrary to me. Using " $d(\text{hypocotyl length})/dT$ " provides a more realistic assessment of the effect sizes in my opinion. Also, I preferred Fig. 2d of the previous version even if now it would include an additional genotype, since it allowed for easier comparison of the effects of sucrose treatment on temperature-dependent hypocotyl elongation.

I also noted a few peculiarities concerning data/statistics:

- Fig. 1c: the distribution of the data points and the descriptive statistics of the hypocotyl lengths in the right panel differ from the previous version of the manuscript even though the data in the left panel are identical to the previous version. This results in discrepancies between the data shown in the right panel and the left panel, even though the data presented are derived from the same experiment, as I understand, and thus should be identical. See for example data for YHBg at 12°C left panel vs. right panel. Please clarify.
- Fig. 2d, +Suc panel: the hypocotyl length data for *hmr-5* shown in the figure do not match the data given in the excel file nor are they in line with the description in the main text or the previous version of the manuscript. Please check again.
- Fig. 2f: even though the starch and sucrose data appear to be the same compared to the previous version of the

manuscript, and the statistical test that has been applied is the same according to the figure captions, the presented p-values are different. Without any explanation for this change provided, I was wondering which one is correct? The same applies to Fig. 5g (It should be noted that this does not affect any of the conclusions drawn based on the data, it just made me wonder.)

- Fig. 5e,f: some of the letters to indicate statistically significant differences are missing.

Minor comments:

- line 44: using the term “embryonic stem” for the hypocotyl seems rather unusual to me. I would suggest to just call it “hypocotyl”.
- line 69: should read “[...] at high temperatures.”
- lines 93-94: I would suggest adding the following reference since in that paper it was shown that plants elongate their hypocotyl in response towards a high temperature treatment that was specifically restricted to daytime: Chung, Balcerowicz et al., 2020, Nat. Plants
- line 108: I find the term “chloroplast-sucrose-mediated” a bit cumbersome and wonder if an alternative would be to stress that high temperatures trigger the degradation of transitory starch in chloroplasts coinciding with an increase in sucrose (as done by the authors anyway) and then just use “sucrose-mediated” instead of “chloroplast-sucrose-mediated” in the following.
- line 263: there is an “o” missing in “hypocotyl”
- lines 365-370: I find these two sentences hard to understand, especially the first one.
- line 372: should read “[...] thermosensing at night because it dictates [...]”
- lines 379-380: I would add: “[...] in chloroplasts and sucrose production in the light.”
- line 400: should read “at high temperatures”

Figure captions:

- Fig 2: delete “ordinary” in front of one-way ANOVA (also applies to Fig. 5); also “multiplicity adjusted” can be removed since Tukey’s HSD test corrects for multiple pairwise comparisons per se. In the caption regarding 2c Ler is missing from the list.

The images or other third party material in this Peer Review File are included in the article’s Creative Commons license, unless indicated otherwise in a credit line to the material. If material is not included in the article’s Creative Commons license and your intended use is not permitted by statutory regulation or exceeds the permitted use, you will need to obtain permission directly from the copyright holder.

Response to Reviewers

We thank both reviewers for their thorough reviews and constructive comments and suggestions. We have made substantial changes to the manuscript based on reviewer comments. We agree with Reviewer #1 that phyB can still play a significant role in thermosensing in the light. What we did was to use a strong light condition ($50 \mu\text{mol m}^{-2} \text{s}^{-1}$ red light) non-permissive for the effect of phyB thermal reversion-dependent thermosensing to show that daytime thermomorphogenesis (the response between 21 and 27 °C) is also gated by two additional phyB-independent thermosensory pathways. We agree with Reviewer #2 that we still do not know precisely the identity of the thermosensor that triggers starch degradation in chloroplasts and that the current data cannot exclude the possibility that this thermosensor resides outside chloroplasts. We added a new figure (Fig. 4) to show how the PIF4 abundance and the expression of PIF4 target genes were changed when starch degradation (*sex1-8*) and sucrose signaling (*tps1-12*) were blocked. Moreover, we revised the model (Fig. 6) to highlight the three major thermosensing pathways and how their roles can be dissected genetically.

Reviewer #1

*The authors examined hypocotyl elongation by varying the red fluence rate and temperature. They concluded that the temperature response could be divided into a phyB-dependent response (12–21°C) and a phyB-independent response (21–27°C). They showed that the phyB-independent temperature response (hypocotyl elongation) is induced by temperature-triggered starch degradation in chloroplasts. This conclusion is supported by the loss of the temperature response in *sex1* and *tps1* mutants and its rescue with sucrose supplementation. The *hmr* mutant, which disrupts both nuclear phyB signaling and chloroplast development, also behaves similarly to the *sex1* and *tps1* mutants. Curiously, however, the authors observed that the *hmr* mutant supplemented with sucrose at 21°C accumulates PIF4 protein at levels comparable to those at 27°C. Yet, the hypocotyl length of the mutant is shorter at 21°C and elongates further when the temperature increases to 27°C. This suggests the presence of another thermosensor that inhibits PIF4 activity. The authors proposed that ELF3 acts as this additional thermosensor. The manuscript provides valuable insights into the complex nature of plant thermoresponses. Below are my comments:*

Temperature Response Classification:

I agree that temperature responses inducing hypocotyl elongation can have multiple facets depending on the temperature. However, dividing the temperature response into “phyB-dependent” and “phyB-independent” categories at 21°C is misleading. The thermal reversion rate of phyB is theoretically proportional to temperature across a broad range, and 21°C is unlikely to serve as a distinct dividing point. It is more plausible that phyB-dependent responses operate throughout the temperature range used in the experiments. The observed “phyB-independent” responses at 21–27°C might be due to the increased dominance of phyB-independent processes at higher temperatures. Terms like “phyB-dominant” and “non-phyB-dominant (?)” might be more appropriate. Additionally, the lack of data points between 21°C and 27°C could contribute to the perception of 21°C as an arbitrary dividing temperature.

Response:

We appreciate the reviewer’s comments. We agree with the reviewer that phyB thermal reversion is expected to play a significant role in temperature-sensing between 10-30 °C (Legris et al. 2016). Our results showed that the phyB-dependent thermosensing (based on the light

repressible temperature response between 12 and 21 °C) plays a significant role in a wide range of light intensities, and it became negligible only when the red light intensity reached $50 \mu\text{mol m}^{-2} \text{s}^{-1}$ (Fig. 1a). The surprising observation was that the temperature responses between 12-21 °C and 21-27 °C showed distinct effects in response to light intensity increases. These results indicate that while phyB thermal reversion plays a primary role in temperature sensing between 12-21 °C, the thermal response between 21-27 °C must also rely on additional phyB-independent thermosensory pathways. Based on the reviewer's suggestion, we named the 21-27 °C response a "multisensor-dependent response".

We designed the experiments using 12, 16, 21, and 27 °C for two reasons. First, we kept the increments between the temperature treatments within 4-to-6 degrees. Second, this design allows the comparison of the commonly studied thermal response, between 21 and 27 °C, to the temperature response in the lower temperature range (12-21 °C).

Experimental Verification using tps mutant:

Since the authors propose that one process is mediated by trehalose-6-phosphate (T6P), the separation of "phyB-dependent" and "phyB-independent" responses across the temperature range could be experimentally clarified by analyzing the fluence- and temperature-dependent hypocotyl lengths of sex1 or tps1 mutants. These mutants are more appropriate than the hmr mutant, as HMR also affects phyB nuclear signaling.

Response:

The goal of the experiments was to use the strong light condition ($50 \mu\text{mol m}^{-2} \text{s}^{-1}$ red light) non-permissive for phyB thermal reversion-mediated thermosensing to dissect additional phyB-independent thermosensing mechanisms between 21 and 27 °C. In other words, the thermosensing mechanisms that operate in $50 \mu\text{mol m}^{-2} \text{s}^{-1}$ red light would be phyB thermal reversion-independent, which include the chloroplast-sucrose-mediated and ELF3-dependent mechanisms.

Under $50 \mu\text{mol m}^{-2} \text{s}^{-1}$ red light, the phyB-only-dependent temperature response between 12 °C and 21 °C was repressed (Fig. 1a). That's the reason why we had to use a lower light intensity ($2.5 \mu\text{mol m}^{-2} \text{s}^{-1}$ red light) as the readout to assess the phyB-dependent response between 12 °C and 21 °C. Although this assay can efficiently identify the effect of phyB thermal reversion, it will not work well in distinguishing the downstream mechanisms because the low light condition ($2.5 \mu\text{mol m}^{-2} \text{s}^{-1}$ red light) works similarly to shade conditions to reduce the activity of phyB and therefore promotes the downstream growth-promoting mechanisms. It has been shown that sucrose signaling also operates under shade conditions (de Wit et al. 2018).

The *hmr* mutant provides a clean no-sucrose background that allowed us to dissect the effect of chloroplast and sucrose in thermosensing (Fig. 6b,c). Although HMR is also required for phyB-mediated degradation of PIF1 and PIF3, this effect is opposite of the sucrose-dependent promotion of PIF4 stabilization. So, despite being slightly taller than Col-0 at 21 °C, the *hmr* mutant is shorter than Col-0 or lack of hypocotyl growth at 27 °C (Qiu et al. 2019). As shown in

this study, the lack of growth and PIF4 accumulation was dependent on the deficiency in chloroplast biogenesis can be rescued by sucrose (Fig. 2c,d and Fig. 3a,b).

Inclusion of phyB and phy Multiple Mutant Data:

The authors' conclusions primarily rely on the temperature responses of wild-type and YHBg lines rather than phyB mutant or phy multiple mutant phenotypes. Since the authors already have access to phy mutants, incorporating this data would strengthen the study. Although interpreting such data could be complex due to the convergence of signaling pathways on PIF4, it would provide additional support for the proposed mechanisms.

Response:

The temperature phenotypes of the *phyB* mutant have been extensively studied previously (Qiu et al. 2019). The *phyB-9* mutant is constitutively tall and insensitive to temperature change. The thermo- and sucrose-inducible PIF4 accumulation was lost in the *phyB-9* mutant, in which PIF4 accumulated to high levels regardless of temperature or sucrose treatment (Fig. 3a). These results indicate that sucrose production promotes PIF4 accumulation by antagonizing phyB-mediated PIF4 degradation. A *phy* multiple mutant would be expected to show the same phenotype as *phyB-9*. So, analyzing *phy* multiple mutants would not provide additional information to the existing model. Instead, we need a mutant specifically abolishing phyB's thermal reversion but not phyB activity, such as YHB, to pinpoint the response that depends specifically on phyB thermal reversion.

Clarification of PhyB Dependency:

The authors noted that sucrose does not increase PIF4 protein levels in the phyB mutant. Does this imply that the "phyB-independent" process is actually phyB-dependent? This could be due to saturation effects. However, without definitive evidence, the authors should be more cautious in drawing conclusions about what is truly phyB-dependent or independent.

Response:

Good question! A major challenge in the field has been to distinguish whether the temperature response is phyB thermosensing-dependent from whether the temperature-dependent hypocotyl elongation response depends on the presence of phyB signaling. The results of this study provide clear evidence to distinguish these two dependencies. As shown in Fig. 6 (the model), the thermo-induced hypocotyl growth response relies on the presence of phyB. Without phyB, neither the chloroplast-sucrose-mediated nor ELF3-dependent mechanism would be functional, and PIF4 would accumulate to high levels and the hypocotyl would be tall at both low and high temperatures (Fig. 3a) (Qiu et al. 2019). The chloroplast-sucrose-mediated thermosensory pathway relies on phyB because it stabilizes PIF4 by antagonizing phyB-mediated PIF4 degradation (although the detailed mechanism is still unclear). The function of ELF3 is phyB-dependent because phyB stabilizes ELF3 presumably by inhibiting COP1-dependent ELF3 degradation (Nieto et al. 2015, 2022). Throughout the study, we used the strong light condition non-permissive for the effects of phyB thermal reversion-dependent

thermosensing to distinguish thermosensing mechanisms that are independent of phyB thermal reversion.

Identities of three thermosensing Systems:

While the authors suggest the presence of three independent thermosensing systems—phyB, starch degradation leading to the T6P, and ELF3—the independence of these systems are not experimentally clear. Authors should experimentally clarify the independence/dependence or should be cautious for their descriptions.

Response:

We revised the manuscript and particularly the model to highlight how the three major thermosensing mechanisms – phyB thermal reversion-dependent, chloroplast-sucrose-mediated, and ELF3-dependent – can be dissected genetically. First, the strong R light condition ($50 \mu\text{mol m}^{-2} \text{s}^{-1}$) can be used as a non-permissive condition for phyB thermal reversion-dependent thermosensing to distinguish the two phyB-independent thermal sensing mechanisms. Notably, the two phyB-independent thermosensing mechanisms play opposing roles in hypocotyl growth. While the chloroplast-sucrose-mediated mechanism promotes hypocotyl growth, the ELF3-dependent mechanism represses hypocotyl growth. As such, high temperatures turns on the chloroplast-sucrose-mediated pathway but represses the ELF3 function. These two mechanisms can be genetically separated using the chloroplast-deficient *hmr* mutant, which lacks the chloroplast-sucrose-mediated thermosensing mechanism, and the *elf3* mutant (Fig. 6).

Reviewer #2:

Key results:

In this manuscript, the authors investigate the molecular mechanisms that gate thermoresponsive hypocotyl elongation, a hallmark of thermomorphogenesis, during daytime, i.e., during the light period, in Arabidopsis. The authors find that warm temperatures induce the remobilisation of transitory starch in the light and that starch degradation is required for thermoresponsive hypocotyl elongation. Furthermore, they show that sucrose supplementation induces the accumulation of PIF4 protein, the central positive transcriptional regulator of thermomorphogenesis, but is not sufficient to induce elongation growth, thereby suggesting the existence of (at least) one additional thermosensor. By comparing the transcriptional responses to sucrose supplementation and to a temperature shift from 21 °C to 27 °C and subsequent mutant analyses the authors suggest that ELF3 acts as the second thermosensor in the light.

Strengths:

The authors nicely dissect temperature-responsive hypocotyl elongation into a phyB-dependent and phyB-independent component using thermal and light fluence gradients, thereby demonstrating the limited contribution of phyB to thermal sensing in the light. The requirement of transitory starch remobilisation in the light for warm temperature-induced hypocotyl elongation shown here suggests a very interesting link between carbon/energy metabolism and temperature-responsive growth that has been studied little in the past, providing several interesting implications for future research. In that sense, the manuscript is of interest to people within the thermomorphogenesis community. However, in my opinion several open questions remain. Also, some of the conclusions drawn based on the data are too strong and not justified as is (see below).

Weaknesses:

(1) The authors repeatedly suggest that chloroplasts act as (or harbour) a thermosensor (e.g., “chloroplasts, as essential thermosensory organelles” (line 338), or “[...] that chloroplasts play an essential role in thermosensing in daylight” (line 186)). However, this is not demonstrated in the manuscript. The *SEX1*-mediated starch degradation in the light at 27 °C that coincides with an increase in endogenous sucrose could also rely on temperature sensing outside of the chloroplasts. Also, the statement in the caption of Fig. 2 “Thermo-inducible starch breakdown in chloroplasts is essential for thermosensing in the light” is too bold in my opinion. I would rather carefully phrase that thermo-inducible starch breakdown is a prerequisite or required for warm temperature-induced elongation growth in the light”.

Response:

We agree with the reviewer that the current data cannot exclude the possibility that a thermosensor outside chloroplasts triggers starch degradation in chloroplasts. We revised the manuscript and refer to this mechanism as a chloroplast-sucrose-mediated thermosensory mechanism.

(2) The authors suggest that “high temperatures induce starch degradation in chloroplasts and subsequent sucrose production and signaling to stabilize PIF4 in daylight” (lines 299ff). However, the experiments presented do not connect starch degradation to PIF4 protein levels directly. Instead, the authors exogenously provided sucrose to different mutant lines and observed increased PIF4 protein levels. Given that the authors demonstrate that warm temperature-induced starch degradation in *Col-0* coincides with enhanced endogenous sucrose levels, and that this is absent in *sex1-8* mutants, I would suggest monitoring PIF4 protein levels in this mutant as a function of temperature in parallel. According to their hypothesis, one would expect reduced PIF4 protein levels in *sex1-8* compared to the wild type. This could also be accompanied by expression analyses of well-characterised PIF4 target genes, e.g., *YUCCA9*, *IAA19*, and would finally establish a link between starch degradation and PIF4 stabilisation.

Response:

We thank the reviewer for the suggestions. We have performed new experiments to show the PIF4 abundance and the expression of select PIF4 target genes in *sex1-8* and *tps1-12*. The new results are included in the new Fig. 3. The results show that, during the 21-to-27 °C transition, PIF4 accumulation was delayed or reduced in *sex1-8* and *tps1-12*, respectively (Fig. 3c,d,f,g). In both *sex1-8* and *tps1-12*, the induction of PIF4 target genes was significantly impaired (Fig. 3e,h). Together, these new results support the conclusion that starch degradation and sucrose signaling promote rapid PIF4 accumulation and the induction of PIF4 targets in response to high temperatures.

(3) Still referring to the warm temperature-induced starch degradation: are there any hypotheses/indications how *SEX1*-mediated starch degradation at 27 °C is induced? Does this rely on transcriptional regulation of *SEX1*, or rather post-translational activation? Is there anything known about the relevance of other starch degrading enzymes at elevated temperatures? I would appreciate a few sentences on this topic in the discussion.

Response:

The thermosensing mechanism leading to starch degradation is unclear. We have started to look at starch-degradation enzymes. However, more in-depth investigations are needed to establish a causality if there is one. In particular, as the reviewer pointed out that we still cannot exclude the possibility of a thermosensor outside chloroplasts. So, we chose not to speculate here. We added the following in the discussion: “The mechanism that triggers starch degradation by high temperatures remains elusive, though similar to dark- and cold-induced starch degradation, thermo-inducible starch degradation also relies on SEX1-mediated starch phosphorylation (Caspar et al. 1991; Lorberth et al. 1998; Zeeman et al. 1998; Yano et al. 2005). Although it is tempting to propose that chloroplasts may be the thermosensor triggering starch degradation, the current data cannot rule out the possibility that starch degradation in chloroplasts is triggered by a thermosensor localized outside chloroplasts.”

(4) In their final model, the authors indicate that the sucrose-dependent effects on PIF4 stability are mediated via T6P (Fig. 5). However, this is not necessarily the case. While it has been shown that elevated T6P levels do indeed enhance PIF4 protein stability in a KIN10-dependent manner (Hwang et al., 2019, EMBO Reports), the effects exerted by sucrose supplementation shown in the manuscript could also be T6P-independent.

In addition, under the conditions used in their experiments (constitutive monochromatic R light), SnRK1 as a regulator of the starvation response, likely is inactive anyway. T6P-dependent stabilisation of PIF4 via repression of SnRK1, however, would require active SnRK1 in the first place to be functional. Therefore, this mechanism does not seem very likely to play a role here, in my opinion. Maybe the authors could discuss this in some more detail. As an additional suggestion, T6P-dependent, GRIK1/SnRK1-mediated effects on PIF4 protein stability (as proposed by Hwang et al., 2019, EMBO Reports) could be indicated as a potential mechanism parallel to T6P-independent effects exerted by increased sucrose levels in the final model (Fig. 5)?

Response:

We thank the reviewer for the insightful thoughts. The new results in Fig. 4 (discussed above) certainly support that T6P plays a major role in promoting PIF4 accumulation. The previous report by Hwang et al. suggested a KIN10-dependent mechanism via SnRK1-dependent phosphorylation of PIF4 (Hwang et al. 2019). We thought that additional experiments would be needed for us to confirm this mechanism. That’s why we did not include SnRK1 in our manuscript. We have largely revised the model (Fig. 6) to highlight the three thermosensing pathways.

(5) Concerning the experiments involving the hmr mutants: I was wondering whether the absence of warm temperature-responsive growth in the hmr mutants (in the absence of exogenous sucrose) could be a result of these seedlings being starving given that genes required for carbon assimilation and sugar production are deregulated in these mutants. Once you provide them with fuel (in the form of exogenous sucrose in this case), they can elongate in response to higher temperature. In this case, I would then argue that in these mutants the starvation response might be negatively impacting on growth thereby preventing warm temperature-responsive hypocotyl elongation. This would be different from the suggested conclusion (lines 166ff) that sucrose production may be required for thermosensing.

Response:

Although sucrose may promote growth as an energy source and a signal, it is the signaling role that triggers thermoresponsive hypocotyl growth, including in *hmr-5*. This conclusion is supported by the following evidence: 1) the sucrose signaling mutant *tps1-12* – which is defective in Tre6P synthesis only – lacks the thermal response despite normal chloroplast development, and the defect in the thermo-inducible hypocotyl growth could not be rescued by exogenous sucrose (Fig. 2c,d); and 2) the hypocotyl growth defect in the chloroplast-deficient mutant, *hmr-5*, is due to a misregulation in phyB-mediated inhibition of hypocotyl growth, as the *hmr-2/phyB-9* double mutant was almost as tall as *phyB-9* (Chen et al. 2010). We had added the above in the Discussion.

Minor Points:

- Lines 136ff.: the authors mention that the phyBY276H variant lacks thermal reversion. I wonder if the lack of thermal reversion has really been demonstrated for phyBY276H (because it is not shown in the two publications cited in line 137), or whether this is extrapolated from the fact that this mutant lacks photoconversion?

Response:

The Y276H substitution created a photochemically inert fluorescent phyB variant. The primary defect of Y276H is a lack of Pr to Pfr transition – i.e., the GAF domain in YHB cannot be photoconverted to a Pfr conformation, as a result the excitation energy would be released by fluorescence – this is how the mutation was isolated (Fischer et al. 2005). So, although YHB is biologically active and the overall conformation is similar to a Pfr, the GAF domain itself is stuck in one conformation and cannot be photoconverted, therefore, also does not have thermal (dark) reversion. These results were confirmed by two other studies (Burgie et al. 2014; Wang et al. 2024).

- Caption Fig. 1b: “Distinct light-intensity effects on phyB-dependent and -independent temperature responses. PhyB-dependent low-temperature response (12-21 °C under dim 2.5 $\mu\text{mol m}^{-2} \text{s}^{-1}$ R light) and phyB-independent high-temperature response (21-27 °C under strong 50 $\mu\text{mol m}^{-2} \text{s}^{-1}$ R light) were quantified as the $d(\text{hypocotyl length})/dT$ derivatives and plotted against light intensity.” What to the underlined specifications mean in this case? The derivatives for the low temperature response and the high temperature response were plotted against the different light intensities. Therefore, specifying specific intensities in the parentheses does not make sense to me.

Response:

We thank the reviewer for the suggestions. We changed the Y-axis to the “percentage hypocotyl increase”. This figure highlights the distinct effects of red light intensity on the phyB dependent response between 12-21 °C and the multisensor-dependent thermal response between 21-27 °C. While increasing light intensity represses the temperature response between 12-21 °C,

it enhances the temperature response between 21-27 °C, indicating that the thermal response is mediated by additional thermosensing mechanisms independent of phyB thermal reversion.

- *Caption Fig. 3e: there is the word “genes” missing between “PIF-induced” and “in hmr-5”.*

Response:

Revised.

The statistical tests applied appear appropriate to me, however, the significance threshold being applied appears rather arbitrary in some instances (e.g., only reporting p values smaller than 0.01 as statistically significant (Fig. 1c and e) is rather uncommon). Is there any particular reason for applying this threshold when doing a single comparison as in Fig. 1c and e?

Response:

We thank the reviewer for the suggestions. The significance threshold should be *p* smaller than 0.05, which was not shown and has now been added.

The figures are of high quality and the data is presented well. Also, the manuscript is clearly written with enough detail in the methods section.

Response:

We thank the reviewer for the comments.

Burgie ES, Bussell AN, Walker JM, Dubiel K, and Vierstra RD. Crystal structure of the photosensing module from a red/far-red light-absorbing plant phytochrome. *Proc Natl Acad Sci U S A.* 2014;**111**(28):10179–10184.

Caspar T, Lin TP, Kakefuda G, Benbow L, Preiss J, and Somerville C. Mutants of *Arabidopsis* with altered regulation of starch degradation. *Plant Physiol.* 1991;**95**(4):1181–1188.

Chen M, Galvão RM, Li M, Burger B, Bugea J, Bolado J, and Chory J. *Arabidopsis* HEMERA/pTAC12 initiates photomorphogenesis by phytochromes. *Cell.* 2010;**141**(7):1230–1240.

Fischer AJ, Rockwell NC, Jang AY, Ernst LA, Waggoner AS, Duan Y, Lei H, and Lagarias JC. Multiple roles of a conserved GAF domain tyrosine residue in cyanobacterial and plant phytochromes. *Biochemistry.* 2005;**44**(46):15203–15215.

Hwang G, Kim S, Cho J-Y, Paik I, Kim J-I, and Oh E. Trehalose-6-phosphate signaling regulates thermoresponsive hypocotyl growth in *Arabidopsis thaliana*. *EMBO Rep.* 2019;**20**(10):e47828.

- Legris M, Klose C, Burgie ES, Rojas CCR, Neme M, Hiltbrunner A, Wigge PA, Schäfer E, Vierstra RD, and Casal JJ.** Phytochrome B integrates light and temperature signals in Arabidopsis. *Science*. 2016;**354**(6314):897–900.
- Lorberth R, Ritte G, Willmitzer L, and Kossmann J.** Inhibition of a starch-granule-bound protein leads to modified starch and repression of cold sweetening. *Nat Biotechnol*. 1998;**16**(5):473–477.
- Nieto C, Catalán P, Luengo LM, Legris M, López-Salmerón V, Davière JM, Casal JJ, Ares S, and Prat S.** COP1 dynamics integrate conflicting seasonal light and thermal cues in the control of Arabidopsis elongation. *Sci Adv*. 2022;**8**(33):eabp8412.
- Nieto C, López-Salmerón V, Davière J-M, and Prat S.** ELF3-PIF4 interaction regulates plant growth independently of the Evening Complex. *Curr Biol*. 2015;**25**(2):187–193.
- Qiu Y, Li M, Kim RJ-A, Moore CM, and Chen M.** Daytime temperature is sensed by phytochrome B in Arabidopsis through a transcriptional activator HEMERA. *Nat Commun*. 2019;**10**(1):140.
- Wang Z, Wang W, Zhao D, Song Y, Lin X, Shen M, Chi C, Xu B, Zhao J, Deng XW, et al.** Light-induced remodeling of phytochrome B enables signal transduction by phytochrome-interacting factor. *Cell*. 2024;**187**(22):6235–6250.e19.
- de Wit M, George GM, Ince YÇ, Dankwa-Egli B, Hersch M, Zeeman SC, and Fankhauser C.** Changes in resource partitioning between and within organs support growth adjustment to neighbor proximity in Brassicaceae seedlings. *Proc Natl Acad Sci U S A*. 2018;**115**(42):E9953–E9961.
- Yano R, Nakamura M, Yoneyama T, and Nishida I.** Starch-related alpha-glucan/water dikinase is involved in the cold-induced development of freezing tolerance in Arabidopsis. *Plant Physiol*. 2005;**138**(2):837–846.
- Zeeman SC, Northrop F, Smith AM, and Rees T.** A starch-accumulating mutant of Arabidopsis thaliana deficient in a chloroplastic starch-hydrolysing enzyme. *Plant J*. 1998;**15**(3):357–365.

Response to Reviewers

We are grateful to Reviewer #2 for their thorough review of our manuscript. We have revised the manuscript and the figures based on the kind suggestions.

Reviewer #2

As stated before, I find the presented study very interesting and agree with reviewer #1 that it provides valuable insights into plant temperature responses. The authors addressed all my concerns, and the requested experimental data has been added. However, I do have a few minor comments.

*The requested data on PIF4 accumulation and downstream target gene expression in *sex1-8* and *tps1-12* have been added and are in line with the hypothesised mechanism. I also appreciate that question whether sucrose serves as energy source or whether it might elicit a kind of “signal” has been discussed in more detail (referring to lines 404-409). Concerning this discussion, I would even add another argument in favour of the signal hypothesis: the fact that sucrose supplementation restores WT-like hypocotyl elongation in *sex1-8* but not in *tps1-12* also points towards the signalling role as crucial for thermoresponsive hypocotyl growth. With hindsight, and with regard to the question as to whether TPS1-dependent T6P signalling is upstream or downstream of starch degradation and sucrose production, it would have been interesting to assess to what extent sucrose supplementation of *sex1-8* and *tps1-12* can restore warm temperature-dependent PIF4 accumulation (referring to Fig. 3c-g) and induction of PIF4 target genes. If the sugar signal was key here, then supplementing *tps1-12* with exogenous sucrose should not be sufficient to restore PIF4 accumulation since the sugar signal is missing, while in *sex1-8* it should be sufficient. But this might be a topic for future investigations.*

Response:

We appreciate the reviewer’s comments. We added the suggested argument in the Discussion section: “While sucrose supplementation restores the thermo-inducible hypocotyl response in *sex1-8*, it did not restore the temperature response in *tps1-12*, indicating that Tre6P signaling mediates thermoresponsive hypocotyl growth downstream of sucrose production.”

The current data clearly demonstrate the genetic relationship between Tre6P and sucrose, which is also consistent with the previous study by Hwang et al. [*EMBO Rep* (2019) 20:e47828]. We agree that future investigations should focus on elucidating the link between Tre6P and PIF4 stabilization.

Concerning the presentation of the hypocotyl elongation data, I preferred the previous presentation as “d(hypocotyl length)/dT” over the “Relative response (%)”. (If this change was done in response to one of my previous comments, I must apologise for this misunderstanding. But this was not what I meant. My previous comment just referred to the figure caption that has been modified accordingly.) Setting the WT response to 100% and calculating the effect sizes of the temperature treatment in the mutant lines relative to WT seems arbitrary to me. Using “d(hypocotyl length)/dT” provides a more realistic assessment of the effect sizes in my opinion. Also, I preferred Fig. 2d of the previous

version even if now it would include an additional genotype, since it allowed for easier comparison of the effects of sucrose treatment on temperature-dependent hypocotyl elongation.

Response:

We have always used the Relative Response in our previous studies on thermomorphogenesis. In the original version of this manuscript, we used “d(hypocotyl length)/dT” to compare the low and high temperature responses. However, during revision, we realized that “d(hypocotyl length)/dT” might not be a good way to characterize dwarf mutants, because the response is not normalized to the hypocotyl length at the lower temperature. So, we decided to switch back to the “Relative Response”, which indicates the percentage of hypocotyl growth triggered by the higher temperature. This change did not alter any conclusions.

We revised the format of Fig. 2d. The original format was to highlight the effect of sucrose within a genotype, but not the comparison of the temperature effects between genotypes. Because the main point is that the sucrose supplement can restore the thermal response in *hmr-5*, *hmr-22*, *sex1-8*, like the wild-type, the current format using the Relative Response is better. Because the data of the same genotype were aligned between the upper and lower panels, the current version can also show the sucrose effect within each genotype.

I also noted a few peculiarities concerning data/statistics:

- Fig. 1c: the distribution of the data points and the descriptive statistics of the hypocotyl lengths in the right panel differ from the previous version of the manuscript even though the data in the left panel are identical to the previous version. This results in discrepancies between the data shown in the right panel and the left panel, even though the data presented are derived from the same experiment, as I understand, and thus should be identical. See for example data for YHBg at 12°C left panel vs. right panel. Please clarify.

Response:

We really appreciate the reviewer for the comments. During revision, we found the variation of the *Ler* samples under $2.5 \mu\text{mol m}^{-2} \text{s}^{-1}$ at 12 °C was unusually high. So, we repeated the experiments of *Ler* and *YHBg* under $2.5 \mu\text{mol m}^{-2} \text{s}^{-1}$ at 12 °C and 21 °C for the right panel of Fig. 1c. We used the new dataset for the revised right panel of Fig. 1c; but, we did not incorporate the new data into the left panel of Fig. 1c. We agree that we should use the same dataset for the left and right panels. In the current version, we remade the left panel. Now, both the left and right panels are from the same dataset.

- Fig. 2d, +Suc panel: the hypocotyl length data for hmr-5 shown in the figure do not match the data given in the excel file nor are they in line with the description in the main text or the previous version of the manuscript. Please check again.

Response:

We appreciate the reviewer's comments. During the reformatting of the figure, the data of *hmr-5* 27 °C with sucrose was switched to the 27 °C without sucrose. We have revised the *hmr-5* 27 °C part of the figure, and now the figure is the same as the previous version.

- Fig. 2f: even though the starch and sucrose data appear to be the same compared to the previous version of the manuscript, and the statistical test that has been applied is the same according to the figure captions, the presented p-values are different. Without any explanation for this change provided, I was wondering which one is correct? The same applies to Fig. 5g (It should be noted that this does not affect any of the conclusions drawn based on the data, it just made me wonder.)

Response:

When we made the Source Data file, we realized that the statistical tests in Fig. 2f and 5g were performed using Welch's t-test (instead of Student's t-test). Because the data are expected to be normally distributed, we changed the analysis to Student's t-test. This is the reason why the p-values in Figs. 2f and 5g are slightly different from the original version. The changes do not alter the conclusions, but we wanted to make sure that the calculations were done correctly.

- Fig. 5e,f: some of the letters to indicate statistically significant differences are missing.

Response:

In the revised version, we modified the format of Fig. 5e-g to highlight the phenotype of *elf3-1* (+s). In Fig. 5e-f, we change the *elf3-1* (+s) to a solid magenta line. During the revision, three letters were missing (as pointed out by the reviewer), which have now been added back. For Fig. 5g, we highlighted the *elf3-1* 21 °C (+s) using magenta color and added the comparison between Col-0 (27 °C) and *elf3-1* 21 °C (+s) to show there is no significant change between them.

Minor comments:

- line 44: using the term "embryonic stem" for the hypocotyl seems rather unusual to me. I would suggest to just call it "hypocotyl".

Response:

We try to use general terms when publishing in a general journal like *Nature Communications*, so we used "embryonic stem" when we described the hypocotyl the first time.

- line 69: should read "[...] at high temperatures."

Response: revised.

- lines 93-94: I would suggest adding the following reference since in that paper it was shown that plants elongate their hypocotyl in response towards a high temperature treatment that was specifically restricted to daytime: Chung, Balcerowicz et al., 2020, *Nat. Plants*

Response:

This is an excellent point. We added the reference.

- line 108: I find the term “chloroplast-sucrose-mediated” a bit cumbersome and wonder if an alternative would be to stress that high temperatures trigger the degradation of transitory starch in chloroplasts coinciding with an increase in sucrose (as done by the authors anyway) and then just use “sucrose-mediated” instead of “chloroplast-sucrose-mediated” in the following.

Response:

We prefer “chloroplast-sucrose-mediated” because chloroplasts are more likely to be or closer to the thermosensor for this pathway. We would like to highlight the requirement of chloroplasts for the thermal response.

- line 263: there is an “o” missing in “hypcotyl”

Response: revised.

- lines 365-370: I find these two sentences hard to understand, especially the first one.

Response:

We revised the two sentences to “This model also explains the dependency of the thermal response on phyB. In the *phyB-9* mutant, the lack of phyB-dependent PIF4 degradation and stabilization of ELF3 resulted in constitutive PIF4 accumulation in both low and high temperatures (Fig. 3a)^{19,62,63}, bypassing the thermal control by the chloroplast-sucrose-mediated and ELF3-mediated pathways (Fig. 6a).”

- line 372: should read “[...] thermosensing at night because it dictates [...]”

Response: revised.

- lines 379-380: I would add: “[...] in chloroplasts and sucrose production in the light.”

Response: revised.

- line 400: should read “at high temperatures”

Response: revised.

Figure captions:

- Fig 2: delete “ordinary” in front of one-way ANOVA (also applies to Fig. 5); also “multiplicity adjusted” can be removed since Tukey’s HSD test corrects for multiple pairwise comparisons per se. In the caption regarding 2c *Ler* is missing from the list.

Response: revised.